# PixFoundation: Are We Heading in the Right Direction with Pixel-level Vision Foundation Models?

## Abstract

Multiple works have emerged to push the boundaries of multi-modal large language models (MLLMs) towards pixel-level understanding. The current trend is to train MLLMs with pixel-level grounding supervision in terms of masks on large-scale labelled data and specialized decoders for the segmentation task. However, we show that such MLLMs when evaluated on recent challenging vision-centric benchmarks, exhibit a weak ability in visual question answering (VQA). Surprisingly, some of these methods even downgrade the grounding ability of MLLMs that were never trained with such pixel-level supervision. In this work, we propose two novel challenging benchmarks with paired evaluation for both VQA and grounding. We demonstrate that simple baselines that are not unified achieve performance that matches or surpasses some of the pixel-level MLLMs. Our paired benchmarks and evaluation enable additional analysis on whether the failure stems from VQA and/or grounding abilities. Furthermore, we propose a paired prompt sensitivity analysis on both the language and visual prompts tailored for the grounding task. Our benchmarking pushes pixel-level MLLMs to improve in both accuracy and robustness, without sacrificing the language abilities. More importantly, we study the research question of "When does grounding emerge in MLLMs with respect to the output tokens?" We propose an interpretability tool that can be plugged into any MLLM to study the aforementioned question. We show that grounding does not necessarily coincide with the exact referring expression in the output, but can coincide with the object parts, its location, appearance, context or state. Code and datasets will be made publicly available.

## 1 Introduction

There have been numerous advancements in pixel-level understanding, including semantic, instance, panoptic segmentation and their video counterpart Zhou et al. (2022); Minaee et al. (2021); Kirillov et al. (2023); Ravi et al. (2024). In addition, significant progress has been made in visual grounding and reasoning Rasheed et al. (2024); Lai et al. (2024), depth estimation Yang et al. (2024) and pixel-level tracking Wang et al. (2023). Visual grounding is tied to relating the visual content with language; examples of its pixel-level tasks include referring segmentation and grounded conversation generation. The majority of these models and tasks have been transformed with the emergence of foundation models Bommasani et al. (2021), especially multi-modal large language models (MLLMs) Liu et al. (2023/); Dai et al. (2023). Nonetheless, pixel-level MLLMs, i.e., MLLMs designed for pixel-level tasks, have shown degradation in their language capabilities Lai et al. (2024).

Recent efforts explored the shortcomings of standard MLLMs in vision-centric benchmarks, Tong et al. (2024b;a). These focused on challenging visual QA tasks that require a form of grounding, emphasizing what these MLLMs fail at, such as counting. Nonetheless, these vision-centric benchmarks did not evaluate the recent pixel-level ones. In this work, we focus on pixel-level visual grounding with a segmentation mask output and propose challenging vision-centric benchmarks that are dedicated to evaluating pixel-level MLLMs, which we call PixMMVP and PixCV-Bench. We then provide a comprehensive paired evaluation for VQA and grounding and analyze the models' robustness to language and visual prompt variations. Our paired evaluation means that the referring segmentation is related to the object of interest in the visual question. Through these, we answer the first research question; *"Are the current pixel-level*

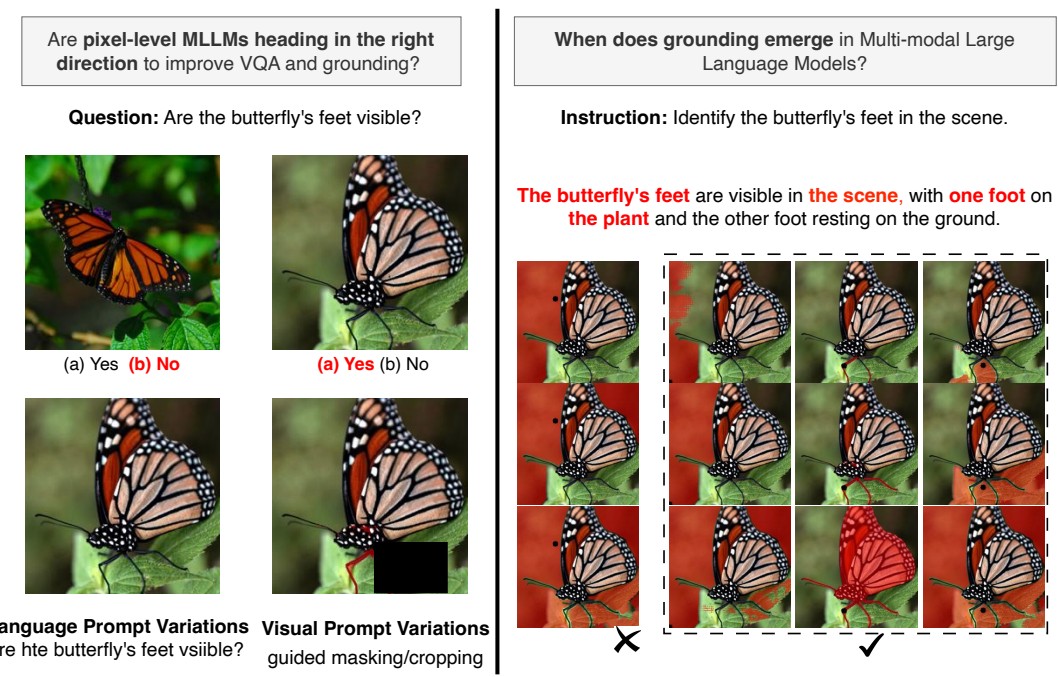

Figure 1: **Research questions we tackle**: (i) the grounding & VQA accuracy and robustness of pixel-level MLLMs in challenging scenarios **(left)**, (ii) when does grounding emerge in standard MLLMs with respect to the output tokens? **(right)**. The latter shows the noun phrases and their corresponding predicted segmentation, highlighted in red. These are extracted from LLaVA 1.5 attention maps with three masks due to the point prompt ambiguity from the maximum attention, highlighted as a black circle.

*MLLMs trained with mask supervision heading in the right direction to improve both grounding and visual question answering (VQA) while maintaining robustness to prompt variations?".* Our findings show that the majority of pixel-level MLLMs still fall short in such a challenging setting. While evidently, some of these show superior performance in grounding, we show that such MLLMs with specialized segmentation decoders in their architecture and with mask supervision can be worse than their counterpart baselines when considering both visual and language abilities. This implies there are still gaps to be explored when building pixel-level MLLMs to ensure they do not lose their language abilities. Moreover, we show some of the well-known general-purpose models with grounding ability, e.g., Qwen2.5-VL Bai et al. (2025), fail in grounding and VQA with language and visual prompt variations. In contrast to previous literature, we contend that evaluating a single task and its robustness in isolation overlooks the mutual influence between language and visual grounding. We believe the current MLLM's direction is towards developing general-purpose models that outperform prior methods across multiple inherently interconnected tasks. Our paired evaluations and prompt sensitivity analysis are a necessary step towards that within the visual grounding literature.

There has been concurrent work Wu et al. (2024) that observed the degradation of pixel-level MLLMs' VQA abilities. Nonetheless, previous efforts used standard evaluation benchmarks that evaluate each task separately. Our benchmarks provide a paired evaluation designed to be vision-centric, with a focus on what MLLMs fall short in and provide the means to interpret the failures of these MLLMs and whether they are stemming from grounding, language capabilities or both. More importantly, unlike concurrent efforts, we focus on the second question of *"When does grounding emerge in MLLMs with respect to the output tokens?"* Our work documents that emerging grounding in MLLMs does not necessarily coincide with the exact language tokens of the referred object, as shown in Fig. 1. We design an interpretability tool that uses the maximum attention point with respect to the output tokens and analyzes the ones that correspond to the best emergent point grounding, using quantitative and qualitative means. While recent works show training-free segmentation emerging from vision language models corresponding to the referred expression Wang et al. (2024); Luo et al. (2024); Hajimiri et al. (2025); Cao et al. (2024), our study quantifies that, within MLLMs, that is not necessarily the case.

In summary, our contributions include: (i) Proposing paired pixel-level vision-centric benchmarks, PixMMVP and PixCV-Bench, with segmentation annotations and referring expression of the object of interest in the corresponding questions. In addition to proposing prompt sensitivity variations tailored to the grounding task. (ii) Benchmarking recent efforts in pixel-level MLLMs where we show that the majority degrade VQA capabilities. More importantly, some of them lag in visual grounding with respect to baselines that are not unified. (iii) We propose a novel interpretability mechanism to study when grounding emerges in MLLMs with respect to the output tokens that benefit from our paired benchmarks. Our mechanism uses the observation that grounding can emerge corresponding to different output tokens describing the object's appearance or location, not necessarily the exact text of the object of interest.

## 2 Related work

**Pixel-level vision foundation models.** There have been various vision foundation models trained for the segmentation task (e.g., SAM and SAM 2.0) Kirillov et al. (2023); Ravi et al. (2024). Orthogonal to this, some methods discussed the ability of vision foundation models such as CLIP and BLIP in image segmentation without any segmentation supervision Luo et al. (2024); Hajimiri et al. (2025); Wang et al. (2024). Yet, they relied on earlier foundation models that did not incorporate the power of large language models. Combining large language models with vision has been extensively researched with pioneering works such as LLaVA Liu et al. (2023/; 2024) and instruct-BLIP Dai et al. (2023).

Multiple works afterwards focused on pixel-level visual grounding in these MLLMs with mask supervision and specialized segmentation decoders Lai et al. (2024); Rasheed et al. (2024); Zhang et al. (2024a;b). However, these methods were lagging in their chat performance. Notably, pixel-level MLLMs were not evaluated on the challenging benchmarks that focused on the shortcomings of MLLMs Tong et al. (2024b;a). Hence, it is still unclear if mask supervision helped to improve their grounding ability on these challenging tasks. In this work, we focus on the previous question to have a better understanding of their performance. While concurrent work Wang et al. (2025) designed better pixel-level MLLMs that are capable of strong VQA and grounding, we study it in relation to its baseline standard MLLM that was not trained with mask supervision and compare their robustness within a language and visual prompt sensitivity evaluation.

**Benchmarking multi-modal large language models.** There is an abundance of standard benchmarks used for evaluating MLLMs (e.g., MMU Yue et al. (2024)) and visual grounding benchmarks (e.g., ref-COCO/+/g Yu et al. (2016); Kazemzadeh et al. (2014)). These have pushed the limits on MLLMs capabilities in terms of VQA and visual grounding. Nonetheless, there have been various works that discussed the shortcomings of MLLMs. One of them discussed the shortcomings in CLIP Radford et al. (2021), which is used in various MLLMs as a visual backbone. They proposed a benchmark, MMVP Tong et al. (2024b), that is focused on the visual aspects within a VQA task. More recently, CV-Bench Tong et al. (2024a) focused on two major tasks that are vision-focused, which are counting and relative positioning. Both were proposed to evaluate MLLMs that do not have the ability to generate segmentation output. Nonetheless, they provide quite challenging scenarios that can act as a strong benchmark for pixel-level MLLMs.

In this work, we extend these two benchmarks with segmentation annotations and referring expressions that correspond to the object of interest within the VQA task, and propose a paired evaluation metric. Moreover, we evaluate these models within a prompt sensitivity evaluation, following previous works Schmalfuss et al. (2025); Chatterjee et al. (2024). We further introduce novel visual prompt variations specifically designed for the grounding task, pushing the limits of pixel-level MLLMs.

**Interpretability of multi-modal large language models.** Explainability in computer vision research falls under two major paradigms, which are: (i) post-hoc methods that analyze already trained black-box models, and (ii) methods that are interpretable by design. In this work, we focus on post-hoc methods that are widely adopted in the literature. Note that the majority of the literature uses explainability and interpretability interchangeably Wan et al. (2026); we similarly refer to our work under interpretability. In multimodal learning, multiple methods focused on the interpretability of vision transformers Xu et al. (2022); Kim et al. (2024) through token or embedding analysis. Studies focused on MLLMs have emerged relying on visual token analysis or chain-of-thought reasoning Gao et al. (2025); Xu et al. (2025). Our work proposes a method that inspects MLLMs to understand when visual grounding occurs with respect to the output

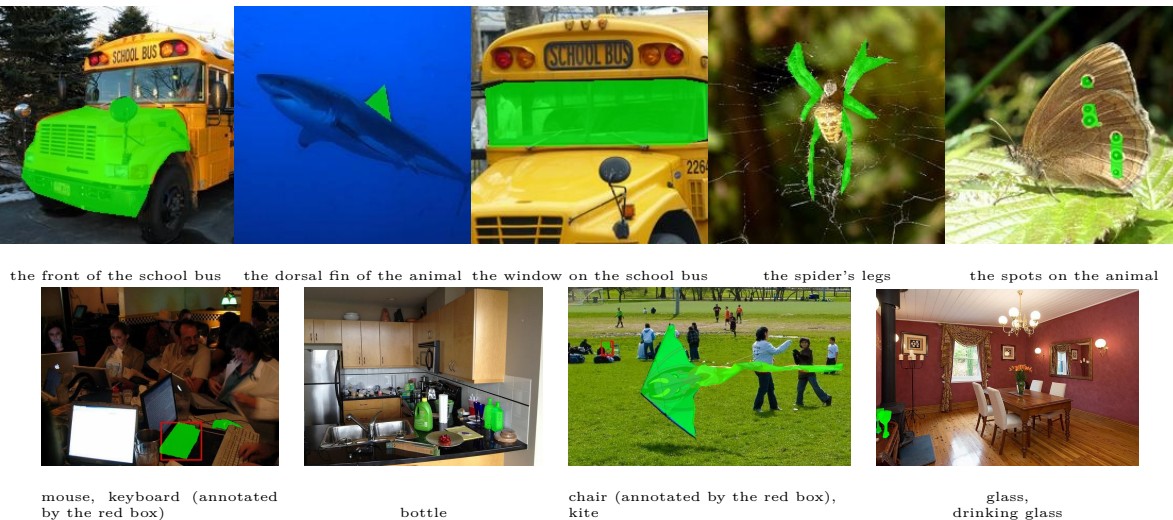

Figure 2: **Examples of ground-truth annotations** for referring expressions in the respective object of interest in the question and their segmentation masks. **First row:** PixMMVP examples, **Second row:** PixCV-Bench examples. Ground-truth highlighted in green.

textual tokens, where we show grounding emerging with tokens describing the objects' location, context or other properties.

# 3 Method and benchmarks

In this section, we describe our two benchmarks and probing techniques for pixel-level MLLMs and MLLMs that were not trained with mask supervision. We focus on the referring segmentation as a representative task of visual grounding. Furthermore, we use visual question answering as a representative task of their vision-language capabilities, with emphasis on the language. Furthermore, we detail the prompt sensitivity for both language and visual input that pushes the limits on MLLMs for both tasks. Finally, we describe our proposed interpretability mechanism that we use to derive a significant insight on MLLMs' ability for visual grounding.

## 3.1 Paired benchmarks for VQA and grounding

**PixMMVP benchmark.** We build upon the recently released MMVP Tong et al. (2024b), which identified clip blind pairs and used them to build a challenging benchmark with the corresponding questions and choices for 300 images. We manually annotate each question with the corresponding object of interest referring expression. There are seven questions only that are not designed to inquire about a specific object in the scene, which are excluded, such as questions inquiring about the view direction of the camera. The referring expressions in our dataset correspond to what fine-grained parts need to be grounded in the image to answer the question. Afterwards, we manually label these objects of interest with polygonal annotations using the VGG annotator Dutta et al. (2016). Hence, we create the first paired benchmark for both VQA and pixel-level visual grounding.

**PixCV-Bench benchmark.** For this benchmark we build upon the 2D component of the recently released CV-Bench Tong et al. (2024a). We specifically select the 2D component, since they are sourced from segmentation datasets (i.e., ADE20K Zhou et al. (2017) and COCO Lin et al. (2014)), which can be used in our proposed benchmark. However, the publicly released CV-Bench does not identify the objects in question and their corresponding segmentation. As such, we use GPT-4o to parse the questions and identify the objects of interest automatically, followed by manual inspection and correction. This provides us with the categories per question that highlight the objects of interest. While seemingly these are categorical annotations, not referring expressions, certain scenarios in CV-Bench are different. Specifically, in the relative positioning

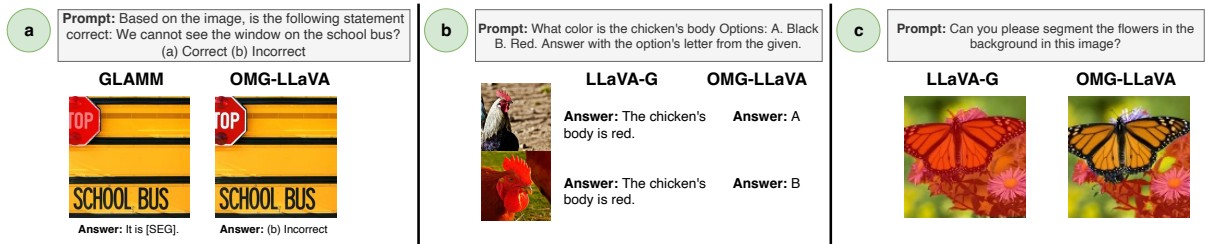

Figure 3: **Failures of pixel-level MLLMs.** (a) The first failure is the degraded performance in visual question answering in some of these models. (b) The second which relates to the first, is the degraded performance in instruction following, where the question is instructing the model to generate one letter from the options but fails to do so. (c) The third is the degraded performance in pixel-level visual grounding in some of these models. The predicted masks corresponding to the [SEG] token/s are highlighted in red.

task, all the questions that include an object highlighted by a red box in the image are annotated with the referring expression, "(annotated by the red box)", beyond simple categories.

Afterwards, we use the expressions from GPT-4o to retrieve the corresponding segmentation mask per image. Furthermore, we use a custom annotation tool to manually filter the objects in question, e.g. selecting only the object mask annotated by the red box and filtering out other instances. Another example that needs manual filtration is when the class in question is a broader category than what is inquired about. Moreover, we identify missing annotations and manually annotate these missing objects. We provide the final paired PixCV-Bench with referring expressions, their segmentation annotations, visual questions and corresponding answers.

Figure 2 shows visual examples with annotations from our benchmarks. We also provide additional analysis to study the out-of-distribution characteristic of PixMMVP and PixCV-Bench (ADE20K) with respect to widely adopted datasets used in training MLLMs for visual grounding in Appendix C.1.

## 3.2 Pixel-level MLLMs study

We utilize the two proposed benchmarks, PixMMVP and PixCV-Bench, and perform an additional prompt sensitivity analysis to evaluate and inspect the failures of these pixel-level MLLMs. The goal is to push these models in the direction of learning visual grounding abilities on the pixel-level without sacrificing the language abilities that are integral to grounding.

**Pixel-level MLLMs failures.** We highlight the failures of the current state-of-the-art pixel-level MLLMs through three probing techniques. First, we highlight the degraded performance in VQA from most of these MLLMs that are trained with mask supervision. We use the following prompt, *"<QUESTION>? <OPTION1> <OPTION2>..."*, as shown in Fig. 3a. Notably, the worst models in this task, e.g., GLAMM Rasheed et al. (2024), are not able to provide an answer and rather refer to a segmentation mask. On the other hand, OMG-LLaVA Zhang et al. (2024b) tries to tackle the question, even if incorrectly.

The second failure we discuss is that these MLLMs exhibit a degraded ability to follow instructions. In order to probe this, we use the following prompt: *"<QUESTION>? a.<OPTION1> b.<OPTION2> ... Answer with the option's letter from the given."* Figure 3b shows an example from one of the worst models in this aspect which is LLaVA-G Zhang et al. (2024a) that is incapable of following the instruction, yet tries to tackle the question in free-form. On the other hand, OMG-LLaVA shows better ability to follow the instructions and answer the question.

Third, we highlight their degraded ability to visually ground objects. Surprisingly, although they were trained with mask supervision for grounding, not all of these models show superior grounding performance on our benchmarks. Figure 3c shows the prompt to generate a segmentation mask for the queried referring expression. The purpose of this probing is to understand whether the failure in these models is purely in its

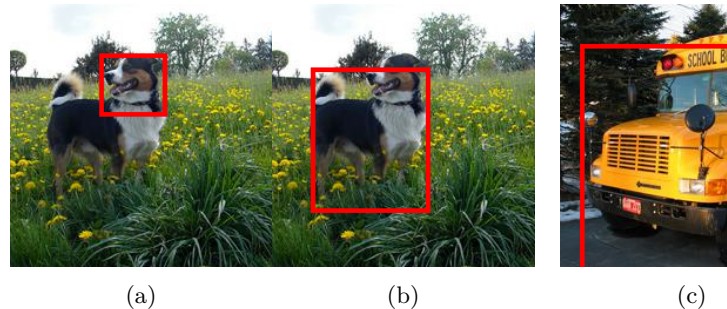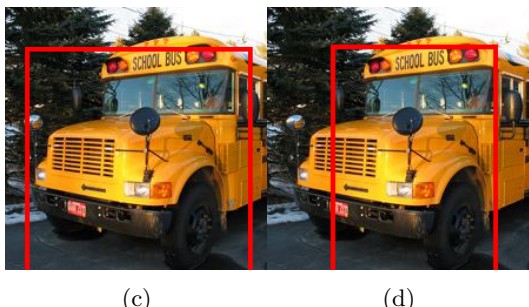

|       (a)       |       (b)       |       (c)       |       (d)       |

Figure 4: **Prompt sensitivity in visual grounding.** Two examples showing the prompt sensitivity in relation to grounding, emphasizing the importance of language in visual grounding. The examples are generated using Qwen2.5-VL. Prompt is (a) "Locate the dog's face and output all the coordinates in JSON format.", (b)"Locate the dog's face, output its bbox coordinates using JSON format.", (c) "Identify the front of the school bus [x1, y1, x2, y2] in JSON format.", (d)"Locate the front of the school bus then output this box coordinates using JSON format.".

language capabilities, or its inability to ground the objects of interest in the corresponding question or both. Figure 3c shows LLaVA-G and OMG-LLaVA, where the latter shows better performance.

**Prompt sensitivity evaluation.** We propose language and visual prompts variations tailored to our paired benchmark to measure the sensitivity of both pixel-level and standard MLLMs. We argue that language is an integral component in the visual grounding task and, consequently, better language abilities in MLLMs can entail better grounding. Figure 4 shows Qwen2.5-VL failures from language prompt variations, where for the same image and referring expression, we modify the language prompt instructing the model for grounding, introducing either a minimal grammatical mistake or a rephrase. It clearly shows the impact of these slight variations on one of the state-of-the-art MLLMs with visual grounding capabilities. This motivates our paired evaluation study, which is augmented with a prompt sensitivity evaluation. We conduct the prompt sensitivity evaluation on PixMMVP, since the dataset is considered out-of-distribution with respect to what the majority of MLLMs were instruction-tuned on, unlike PixCV-Bench (COCO).

For the language variations, we follow previous works Chatterjee et al. (2024) that rely on three variants. These include: (i) spelling errors from omission, transposition, and addition of letters that are randomly generated across random locations, (ii) random selection of the template, and (iii) paraphrasing using GPT-4o. We conduct these variations across both the VQA and the visual grounding task. For the VQA task, we generate eight random locations in both the question and the choices to change, excluding the enumeration of the choices. Similarly, we randomly select eight locations in the instruction, prompting the model to generate an option's letter. Table 1 shows the prompt templates that are used in the language prompt sensitivity evaluation in the VQA and the visual grounding. For the visual prompts variations, we propose novel variations that are tailored for visual grounding through guided random cropping and masking. Assume the bounding box for the ground truth mask with its top left coordinates, width, and height as $[x, y, w, h]$. We randomly generate the parameter, $P_c \in [0.5, 0.8]$, to crop the region of interest, $[0, 0, P_c.w, P_c.h]$. Similarly, we randomize the parameter, $P_m \in [0.5, 0.9]$ and mask out the region of interest with a black rectangle as, $[\bar{x}, \bar{y}, P_m.w, P_m.h]$, where $\bar{x} \in [x, x + \frac{w}{2}], \bar{y} \in [y, y + \frac{h}{2}]$, are randomly generated. Figure 5 shows MLLMs' original and degraded performance under these two visual variations across the two control parameters, $P_c, P_m$, as a motivation.

### 3.3 Interpretability mechanism

In addition to evaluating state-of-the-art pixel-level MLLMs, we propose a novel interpretability mechanism to study when grounding emerges in MLLMs with respect to the output tokens. We are inspired by a concurrent work Cao et al. (2024) that identified the emergent pixel-level grounding in MLLMs without the need for any mask supervision. Specifically, we use their attend and segment meta architecture as one of our baselines. However, we are the first to discuss when such grounding emerges in these models. We identify

| # | Prompt Template |
|---|---|
| 1 | Ques:QUES\n|CHOICES \nInstruct:|INSTRUCT\nAns: |
| 2 | QUES \n| CHOICES \n| INSTRUCT \nAnswer: |
| 3 | QUES | CHOICES | INSTRUCT |
| 4 | Question-QUES | CHOICES. | INSTRUCT. Answer- |
| 5 | QUES \t| CHOICES \t| INSTRUCT Answer:: |
| 6 | Question, QUES | CHOICES. | INSTRUCT. Answer, |
| 7 | Q: QUES \n| CHOICES \n| INSTRUCT A: |
| 8 | QUES \t| CHOICES \t| INSTRUCT A: |
| 9 | QUES | CHOICES | INSTRUCT |
| 10 | Q::QUES | CHOICES \n| INSTRUCT \n A:: |

| # | Prompt Template |
|---|---|
| 1 | Can you please segment/detect EXPR in the given image |
| 2 | Can you segment/detect EXPR in this image? |
| 3 | Can you identify EXPR in this image? (with grounding) |
| 4 | Identify EXPR in the scene, with grounding. |
| 5 | Locate the EXPR and output a tight mask/box. If the object does not exist in the image don't generate any masks/boxes. |
| 6 | Output mask/box for the EXPR |

Table 1: **Language prompt sensitivity. Left:** templates used in the VQA. QUES, CHOICES, INSTRUCT correspond to the question, choices and the instruction to generate an option's letter, resp. **Right:** templates used in the visual grounding. According to the model's output, the prompt either uses (segment, mask, masks), or (detect, box, boxes) in the prompt. EXPR corresponds to the referring expression.

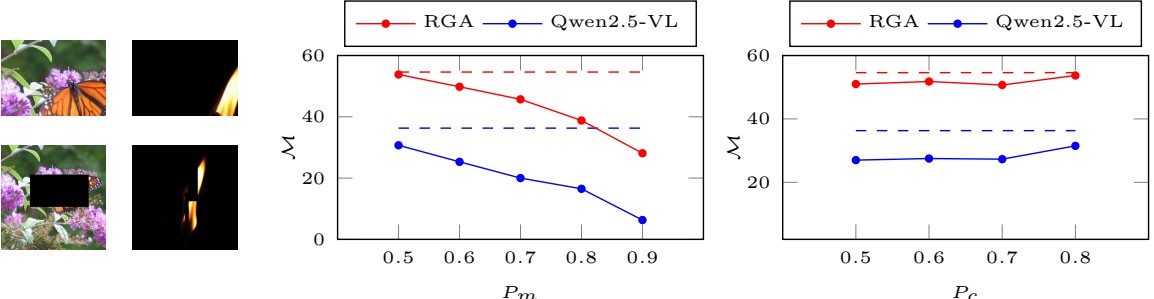

Figure 5: **Visual prompt sensitivity.** It shows the best two MLLMs in visual grounding sensitive to perturbations in the visual prompt using: **Left:** guided masking and **Right:** guided cropping, to motivate this study. The original performance of these MLLMs is presented as an upper bound (dashed line). It also shows example prompts with these variations.

an interesting connection between the identified output tokens and the output grounding from the attention maps. This gives insights into how these models reason.

We extract the raw attention map for the $i^{th}$ output token, $A_i \in [0,1]^{n_{\text{layer}} \times n_{\text{head}} \times (x+hw+y+i-1)}$, where $n_{\text{layer}}, n_{\text{head}}$ are the number of layers and heads, resp. Then, $x, y$ are the number of input language tokens before and after the visual tokens, respectively, while $hw$ are the height and width of the input image features. Only the attention corresponding to the visual tokens of length $hw$ is used, and these attention maps are averaged across the layers and heads, resulting in $\bar{A}_i \in [0,1]^{h \times w}$. This is further normalized across all the output, $\tilde{A}_i = \bar{A}_i - \frac{1}{N}\sum_{j=1}^{N} \bar{A}_j$ for $N$ output tokens. The attend and segment baseline depends on using the spaCy natural language processing tool Honnibal et al. (2020) to identify the noun phrases and associate them with the queried referring expressions. Thus, the spaCy embeddings closest to the queried expression are used to select the $n^{th}$ noun phrase, and its tokens to construct the set of selected token indices, $\mathbb{S}_n$. Their attention maps are averaged to construct the noun phrase attention map, $\tilde{A}_n = \frac{1}{|\mathbb{S}_n|}\sum_{i \in \mathbb{S}_n} \tilde{A}_i$. This is followed by extracting the maximum attention point, $P_n = \arg\max(\tilde{A}_n)$, to feed into SAM Kirillov et al. (2023) as a point prompt.

For our interpretability mechanism, we build upon the previous pipeline and provide an *oracle* upper bound and an *automatic* method, which we call PixFoundation. We propose an additional mask selection that searches for the emergent grounding of the referring expression within the output. For each noun phrase, $n$, we compute its respective point prompt, $P_n$, and the set of its output masks from SAM, $\mathbb{M}_n$, while accounting for point prompt ambiguity. We generate images that highlight each of the masks in this set as $\mathbb{I}_n$ and use the MLLM as a judge to select the highlighted image that best describes the queried expression, $n_s = \mathcal{F}_{\text{Judge}}(\{\mathbb{I}_n\}_{n=1}^N)$. In this case, the selected $n_s^{th}$ noun phrase is not necessarily the closest to the queried expression, because our tool allows the MLLM to search across all the noun phrases. Furthermore, we consider that the point prompt ambiguity is crucial in identifying fine-grained details; as such, we allow for our mask selection to consider three masks from SAM for each point and its respective noun phrase.

| Image | Referring Expression | Concept Category | Noun Phrase | Output |
|-------|---------------------|------------------|-------------|--------|
| 1 | the butterfly's wings | Color & Appearance | orange wings | In the image, there is a butterfly with orange wings. |
| 3 | the flame of the match | Location & Position | the top | The flame of the match is located at the top of the image, surrounded by darkness. |
| 6 | the dog's face | Color & Appearance | a black and white dog | The dog's face in the scene is a black and white dog with a black nose. |
| 161 | the minute hand of the clock | Location & Position | the 12 o'clock position | The minute hand of the clock in the scene is located at the 12 o'clock position. |

1   3   6   161

Figure 6: **Our Interpretability mechanism** showing concept categories where the grounding emerges in PixMMVP using LLaVA 1.5 (7B). **Top:** referring expression, output response, noun phrases and concepts corresponding to the grounding using the *oracle* selection. **Bottom:** the four images with predicted segmentation mask, highlighted in red, using the *oracle* selection and max. attention, highlighted as a black circle. It shows the referring segmentation emerging in different noun phrases than the queried expression.

The *automatic* method can be used with either closed-source or open-source MLLMs without requiring groundtruth segmentation. It can be used as an interpretability tool to have a better understanding of where MLLMs look and when the correct grounding occurs with respect to the output tokens. As shown in Sec. 4.2 our insight is significant and validated with considerable gains up to 30%, surpassing the attend and segment baseline. In the *oracle*, we rely on the ground-truth mask to select the correct token and its corresponding segmentation with the highest intersection over union as an upper bound.

Figure 6 shows qualitative examples from our interpretability tool, the oracle variant, applied to LLaVA 1.5, where the best masks and their respective noun phrases do not necessarily correspond to the queried referring expression. We also study the concept categories for the noun phrases where the best grounding occurs, relying on our interpretability tool, refer to Sec. 4.3. We show how the output from our interpretability mechanism and the oracle variant is better capable of identifying the object part, "dorsal fin", from LLaVA 1.5 without being trained with mask supervision, unlike other pixel-level MLLMs, while still maintaining its language capabilities. Considering that pixel-level MLLMs rely on standard ones as their base MLLM, our benchmarking and interpretability study questions whether there could be better pixel-level MLLMs built without the loss of their original language and grounding capabilities.

# 4 Experiments

## 4.1 Experimental setup

**Evaluation benchmarks, protocols and metrics.** PixMMVP is composed of 300 images paired with questions, choices, referring expressions and segmentation masks, while PixCV-Bench has 1,438 images with their corresponding annotations similarly. On each benchmark, we evaluate the VQA and referring segmentation following three probing techniques and report their metrics. The first probing is to evaluate the VQA ability, where the accuracy is computed using GPT-4o following Tong et al. (2024b) as, $\mathcal{A}\dagger$. If the model generates a segmentation without explicitly asking it to, it is evaluated with respect to the ground-truth referring segmentation in terms of mean intersection over union as $\mathcal{M}\dagger$. The second probing prompts the model to identify the referred expression and evaluates the mean intersection over union reported as $\mathcal{M}$.

The third probing following Tong et al. (2024a) instructs the model to generate an option letter and evaluate the accuracy directly without GPT-4o, reported as $\mathcal{A}$. We evaluate the score of each model, $\mathcal{S}$, which is the harmonic mean across both referring segmentation and VQA,

$$\mathcal{S} = \frac{2}{\frac{1}{\max(\mathcal{A},\mathcal{A}\dagger)} + \frac{1}{\max(\mathcal{M},\mathcal{M}\dagger)}}. \tag{1}$$

For the language prompt sensitivity evaluation in VQA task, we randomly generate 10 variations for each of the three language variants, and for the visual one, we generate four for each of the two variants. As such, we have in total 30 variations for the VQA and eight for referring segmentation; for simplicity, we report the mean of the metric used across all the variations. For the VQA accuracy, we follow the third probing, where accuracy is evaluated using the generated option's letter with simple postprocessing on the model's output. To avoid observations from previous work Hua et al. (2025) that prompt sensitivity failures can stem from the evaluation and post-processing itself, we evaluate with GPT-4o the same output and report both, as $\mathcal{A}$ and $\mathcal{A}$ w/ GPT, respectively. For the language prompt sensitivity in visual grounding, we use two variations per prompt template, resulting in 12 variations for the prompt without the referring expression being impacted. The language variations randomly modify eight characters in the grounding prompt, where modifications can be either omission, addition or transposition. We also perform separate variations on the referred expression only within the aforementioned three modifications.

**Compared methods.** We focus on evaluating five state-of-the-art pixel-level MLLMs; LISA Lai et al. (2024), GLAMM Rasheed et al. (2024), OMG-LLaVA Zhang et al. (2024b), LLaVA-G Zhang et al. (2024a) and RGA Wang et al. (2025). These were either equipped with a SAM or SAM 2 decoder or with other powerful decoders, e.g., OMG-Seg. Some of these were trained with the SAM dataset, SA-1B, extended to the visual grounding Rasheed et al. (2024) or other large-scale segmentation data. Furthermore, we evaluate the attend and segment (a+s) Cao et al. (2024), the *oracle* relying on the highest intersection over union in mask selection (PixFoundation†), the *automatic* selection (PixFoundation) and a simple baseline that uses the output bounding box/point to prompt SAM without any selection. These are implemented on top of four base MLLMs, which are LLaVA 1.5 (7B, 13B) Liu et al. (2024), Cambrian (8B) Tong et al. (2024a) and Qwen2.5-VL Bai et al. (2025). The automatic selection is implemented using GPT-5.1. We initially identify the images that do not have the referring expression, and correspondingly, output an empty mask. Specifically, we prompt the models with the question: "Does this image have <EXP>? Answer with Yes/No only." We subsequently employ an iterative prompting strategy in which the MLLM is shown $N$ images with the predicted segmentations highlighted and is tasked with selecting the best candidate. The final iteration performs a selection among the candidates retained from previous iterations. The number of images $N$ is determined based on the MLLM's constraints on the context length. Each iteration, we use the prompt, "Select the image that has <EXP> best highlighted in red colour than the others? Answer with a number from 1 to <N>. Mention the number only." Additional details are in Appendix A.

### 4.2 Are the current pixel-level MLLMs heading in the right direction?

In order to answer this, we evaluate each of these pixel-level MLLMs' capabilities in VQA and referring segmentation in challenging tasks. Table 2 shows the results on the challenging PixMMVP and the respective prompt sensitivity ones. From the accuracy of VQA, standard MLLMs that are not trained for mask output surpass their pixel-level counterpart with 3-14%. This is especially evident in the language prompt sensitivity, where RGA, the best pixel-level MLLM, drops VQA accuracy to 28.5%, which is worse than other standard MLLMs, i.e., Cambrian (8B) and Qwen2.5-VL (7B) at 44.2% and 33.8%, respectively. Furthermore, RGA shows severe failure in $A\dagger$, which is attributed to its sensitivity to the VQA prompt, as it can only answer the multiple-choice question when instructed to generate a letter, while exhibiting limited ability to generate free-form answers. As for the referring segmentation, some of these pixel-level MLLMs, e.g., OMG-LLaVA and LLaVA-G, underperform the baselines.

Comparing PixFoundation and the baseline (a+s) clearly shows that our interpretability tool is better because of the understanding that the grounding does not necessarily coincide with the output noun phrase closest to the referring expression, unlike the (a+s) baseline. This is further confirmed in the point accuracy results

| Method | Venue | Grounding Output | PixMMVP | | | | | PixMMVP Prompt Sensitivity | | | |
|---|---|---|---|---|---|---|---|---|---|---|---|
| | | | $\mathcal{A}$† | $\mathcal{A}$ | $\mathcal{M}$† | $\mathcal{M}$ | $\mathcal{S}$ | $\mathcal{A}$ | $\mathcal{A}$ w/ GPT | $\mathcal{M}$ | $\mathcal{S}$ |
| **Pixel-level MLLMs w/ Mask Output** | | | | | | | | | | | |
| OMG LLaVA (7B) ◊ | NeurIPS 2024 | Mask | 12.0 | 12.0 | 17.8 (13.8) | 38.0 (38.8) | 18.2 (18.3) | 5.9 | 11.8 | 29.7 | 9.8 |
| GLAMM (7B) | CVPR 2024 | Mask | 1.3 | 2.7 | 31.5 (30.6) | 47.4 (48.3) | 5.1 (5.1) | - | - | 41.9 | - |
| LISA (7B) | CVPR 2024 | Mask | 7.3 | - | 18.1 (14.1) | 42.9 (42.9) | 12.5 (12.5) | - | - | 35.0 | - |
| LLaVA-G (7B) | ECCV 2024 | Mask | 9.3 | - | 19.2 (15.2) | 28.8 (29.1) | 14.0 (14.1) | - | - | 26.8 | - |
| RGA (7B) ★ | ICCV 2025 | Mask | 5.3 | 51.3 | - | **52.7 (54.6)** | **52.0 (52.9)** | 23.2 | 28.5 | 45.8 | 35.1 |
| **Standard MLLMs Extended for Mask Output** | | | | | | | | | | | |
| LLaVA 1.5 (7B) + SAM | baseline (ours) | Box2Mask | 27.3 | 28.0 | - | 38.2 (40.1) | 32.3 (33.0) | 18.4 | 18.7 | 21.9 | 20.2 |
| LLaVA 1.5 (13B) + SAM | baseline (ours) | Box2Mask | 39.3 | 30 | - | 45.5 (47.7) | 42.2 (43.1) | 24.4 | 23.7 | 24.9 | 24.6 |
| Cambrian (8B) * + SAM | baseline (ours) | Box2Mask | 52.0 | **52.0** | - | 39.9 (41.8) | 45.2 (46.3) | - | 44.2 | 28.4 | 34.6 |
| Qwen2.5-VL (7B) ★ + SAM | baseline (ours) | Point2Mask | **54.7** | 38.0 | - | 37.6 (36.3) | 44.6 (43.6) | **29.6** | 33.8 | 24.8 | 28.6 |
| Qwen2.5-VL (7B) ★ + SAM | baseline (ours) | Box2Mask | **54.7** | 38.0 | - | 33.2 (31.0) | 41.3 (39.6) | **29.6** | 33.8 | 26.0 | 29.4 |
| | | | | | | | | | | | |
| LLaVA 1.5 (7B) + (a+s) | baseline (Arxiv 2025) | Point2Mask | 27.3 | 28.0 | 11.8 (7.8) | 18.1 (14.1) | 22.0 (18.8) | 18.4 | 18.7 | 16.5 | 17.5 |
| LLaVA 1.5 (13B) + (a+s) | baseline (Arxiv 2025) | Point2Mask | 39.3 | 30 | 12.1 (8.5) | 17.9 (13.9) | 24.6 (20.5) | 24.4 | 23.7 | 16.3 | 19.5 |
| Cambrian (8B) * + (a+s) | baseline (Arxiv 2025) | Point2Mask | 52.0 | **52.0** | 19.5 (15.5) | 16.7 (13.3) | 28.4 (23.9) | - | 44.2 | 18.8 | 26.4 |
| | | | | | | | | | | | |
| LLaVA 1.5 (7B) + PixFoundation | interp. tool (ours) | Point2Mask | 27.3 | 28.0 | 31.2 (29.2) | 42.8 (42.1) | 33.9 (33.6) | 18.4 | 18.7 | 37.9 | 25.0 |
| LLaVA 1.5 (13B) + PixFoundation | interp. tool (ours) | Point2Mask | 39.3 | 30 | 27.1 (26.0) | 41.3 (40.1) | 40.3 (39.7) | 24.4 | 23.7 | 37.5 | 29.6 |
| Cambrian (8B) * + PixFoundation | interp. tool (ours) | Point2Mask | 52.0 | **52.0** | 47.0 (44.7) | 41.8 (41.4) | 49.4 (48.1) | - | 44.2 | 33.6 | **38.2** |
| Qwen2.5-VL (7B) ★ + PixFoundation | interp. tool (ours) | Point2Mask | **54.7** | 38.0 | - | 46.5 (44.2) | 50.3 (48.9) | **29.6** | 33.8 | 31.6 | 32.7 |
| | | | | | | | | | | | |
| LLaVA 1.5 (7B) + PixFoundation† | upper bound (ours) | Point2Mask | 27.3 | 28.0 | **37.8 (34.8)** | **55.3 (53.1)** | **37.2 (36.7)** | 18.4 | 18.7 | **56.4** | **28.1** |
| LLaVA 1.5 (13B) + PixFoundation† | upper bound (ours) | Point2Mask | 39.3 | 30 | **36.8 (33.7)** | **55.2 (53.0)** | **45.9 (45.1)** | 24.4 | 23.7 | **55.2** | **33.8** |
| Cambrian (8B) * + PixFoundation† | upper bound (ours) | Point2Mask | 52.0 | **52.0** | **68.7 (67.1)** | **72.1 (70.8)** | **60.4 (60.0)** | - | 44.2 | **70.9** | **54.5** |
| Qwen2.5-VL (7B) ★ + PixFoundation† | upper bound (ours) | Point2Mask | **54.7** | 38.0 | - | **55.0 (52.8)** | **54.8 (53.7)** | **29.6** | 33.8 | **38.9** | **36.2** |

Table 2: **PixMMVP evaluation.** VQA accuracy using the first and third probing (i.e., $\mathcal{A}$† & $\mathcal{A}$ resp.). Visual grounding w/ mask output using the first two probing (i.e., $\mathcal{M}$† & $\mathcal{M}$ resp.). *,◊,★: models using Llama 3 (8B), InternLM2 (7B), and Qwen2.5 (7B), resp., unlike the rest that are relying on Vicuna (7B and 13B) for the base LLM. - : indicates either the model can not be evaluated in that setting, or has low results below 1%. (.): is the mIoU while discarding all the images that were labelled with no objects of interest. $\mathcal{S}$: denotes the score of the MLLM, which is the harmonic mean of VQA and grounding accuracies. The *oracle* results are highlighted in red, the best and second best are **bolded** and underlined, resp.

in Figure 7a. When looking at the overall score, $\mathcal{S}$, in PixMMVP, our concurrent work (RGA) shows strong performance that is better than its respective model Qwen2.5-VL, yet in the prompt sensitivity analysis, it shows lower performance than Cambrian with PixFoundation. The second-best pixel-level MLLM, OMG-LLaVA, which was prior to the release of our benchmark, consistently lags behind our baselines.

Various factors can impact the language and visual abilities of these MLLMs, especially in visual grounding and VQA. These results could be attributed to training recipes, design choices or datasets; this is evident with RGA performance surpassing other pixel-level MLLMs, which rely on mixing both QA and segmentation datasets during training and expanding them with video tasks. However, even with RGA, its VQA abilities downgraded drastically in the language prompt sensitivity with a factor of 1.8×, while the standard MLLM Cambrian degraded with a factor of 1.2×. It also suffered in the $\mathcal{A}$† metric which shows its reliance on specific instructions and limited ability in free-form text generation. Unlike RGA, Qwen2.5-VL and Cambrian generate grounding output directly in language. There could be potential opportunities to improve the language capabilities through exploring these design choices further in the segmentation domain. Additional results and discussion on this is provided in Appendix B.4. Table 3 shows PixCV-Bench results. Our concurrent work, RGA, shows strong performance, followed closely by Qwen2.5-VL. It also shows that Point2Mask is worse than Box2Mask in our simple baselines in PixCVBench, which could be due to the referring expressions inquiring about multiple objects/instances more than in PixMMVP due to the nature of the questions, e.g., relative positioning. Furthermore, Point2Mask shows higher sensitivity to visual prompt variations than Box2Mask.

Our baselines rely on the maximum attention per output noun phrase to prompt SAM for the segmentation mask. Nonetheless, as a lower bound analysis, we evaluate the performance if we use a random point as a prompt instead. For fair comparison, we generate random points with the count of output masks that the oracle has to select among (i.e., the number of the output noun phrases). We conduct this ablation on PixMMVP using LLaVA 1.5 (7B) base MLLM, with random point prompts followed by the *oracle* selection among their SAM masks. Figure 7c shows that random + oracle lags behind the correct one using the maximum point (i.e., 1st) with around 28.9%. More importantly, we confirm the stability of the results if we select the second-best or third maximum attention (i.e., 2nd or 3rd), which are on par with the maximum

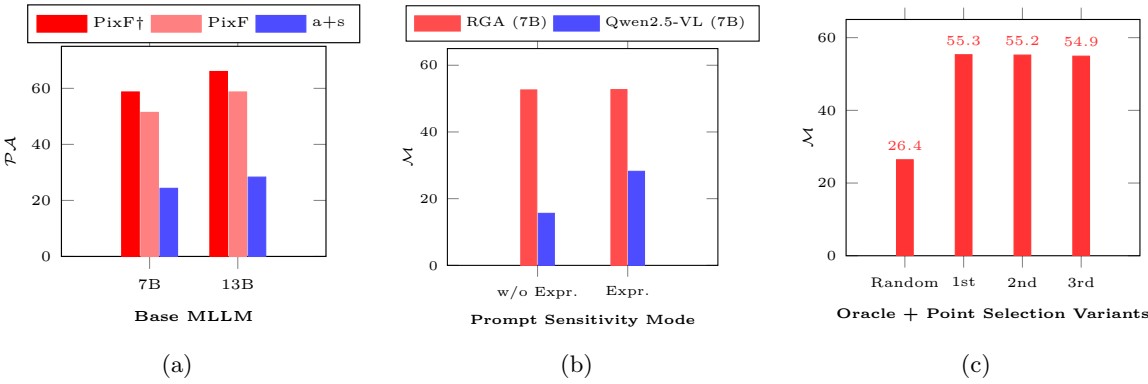

(a)        (b)        (c)

Figure 7: **Ablations on PixMMVP benchmark.** (a) Point accuracy ($\mathcal{PA}$) ablation of LLaVA 1.5 (7B & 13B), PixF: PixFoundation. It shows consistently that our PixFoundation automatic and oracle variants surpass the attend and segment method, confirming that visual grounding occurs in noun phrases that are not necessarily the closest to the referring expression. (b) Language prompt sensitivity in the visual grounding task of RGA and Qwen2.5-VL, showing two modes (w/o Expr: variations occur to the prompt without the referring expressions, Expr: variations occur only to the referring expressions), using intersection over union ($\mathcal{M}$). It shows how state-of-the-art model Qwen2.5-VL is severely impacted by variations in the language prompt, impacting its visual grounding ability. (c) Random baseline vs. maximum attention point, second maximum and third maximum ablation using intersection over union ($\mathcal{M}$). It shows the stability of our results regardless of using the first, second or third maximum and confirms its superiority with respect to a random baseline variant. All variants are evaluated within the oracle selection.

| Method | Grounding Output | PixCV-Bench | | | |
|---|---|---|---|---|---|
| | | $\mathcal{A}\dagger$ | $\mathcal{A}$ | $\mathcal{M}$ | $\mathcal{S}$ |
| **Pixel-level MLLMs w/ Mask Output** | | | | | |
| OMG LLaVA (7B) ◇ | Mask | 12.0 | 42.1 | 50.5 | 45.9 |
| GLAMM (7B) | Mask | - | - | **51.9** | - |
| LISA (7B) | Mask | 3.7 | - | 48.1 | 6.7 |
| LLaVA-G (7B) | Mask | 14.1 | 4.4 | 17.6 | 15.8 |
| RGA (7B) ⋆ | Mask | - | **72.6** | 51.1 | **56.0** |
| **Standard MLLMs Extended for Mask Output** | | | | | |
| LLaVA 1.5 (7B) + SAM | Box2Mask | 17.4 | 60.3 | 29.8 | 39.9 |
| LLaVA 1.5 (13B) + SAM | Box2Mask | 14.5 | 61.4 | 32.9 | 42.8 |
| Cambrian (8B) ∗ + SAM | Box2Mask | **62.2** | 72.2 | 34.2 | 46.4 |
| Qwen2.5-VL (7B) ⋆ + SAM | Point2Mask | 48.9 | 60.8 | 33.7 | 43.4 |
| Qwen2.5-VL (7B) ⋆ + SAM | Box2Mask | 48.9 | 60.8 | 48.9 | 54.2 |
| | | | | | |
| LLaVA 1.5 (7B) + (a+s) | Point2Mask | 17.4 | 60.3 | 15.7 | 24.9 |
| LLaVA 1.5 (13B) + (a+s) | Point2Mask | 14.5 | 61.4 | 14.9 | 24.0 |
| Cambrian (8B) ∗ + (a+s) | Point2Mask | **62.2** | 72.2 | 15.9 | 26.1 |
| | | | | | |
| LLaVA 1.5 (7B) + PixFoundation | Point2Mask | 17.4 | 60.3 | 34.1 | 43.6 |
| LLaVA 1.5 (13B) + PixFoundation | Point2Mask | 14.5 | 61.4 | 33.9 | 43.7 |
| Cambrian (8B) ∗ + PixFoundation | Point2Mask | **62.2** | 72.2 | 38.4 | 50.1 |
| | | | | | |
| LLaVA 1.5 (7B) + PixFoundation† | Point2Mask | 17.4 | 60.3 | 49.7 | 54.5 |
| LLaVA 1.5 (13B) + PixFoundation† | Point2Mask | 14.5 | 61.4 | 50.6 | 55.5 |
| Cambrian (8B) ∗ + PixFoundation† | Point2Mask | **62.2** | 72.2 | 64.4 | 68.1 |
| Qwen2.5-VL (7B) ⋆ + PixFoundation† | Point2Mask | 48.9 | 60.8 | 43.6 | 50.8 |

Table 3: **PixCV-Bench evaluation.** VQA accuracy using the first and third probing (i.e., $\mathcal{A}\dagger$ & $\mathcal{A}$ resp.). Visual grounding w/ mask output using mIoU in the second probing (i.e., $\mathcal{M}$).

attention (i.e., 1st). Figure 8 shows further qualitative ablation supporting that our PixFoundation† is better capable of grounding fine-grained details because of a better understanding of when grounding emerges.

We evaluate the best pixel-level MLLM and its base MLLLM for the language prompt sensitivity in the visual grounding task, aside from the VQA task. Figure 7b compares RGA with its baseline MLLM, Qwen2.5-VL,

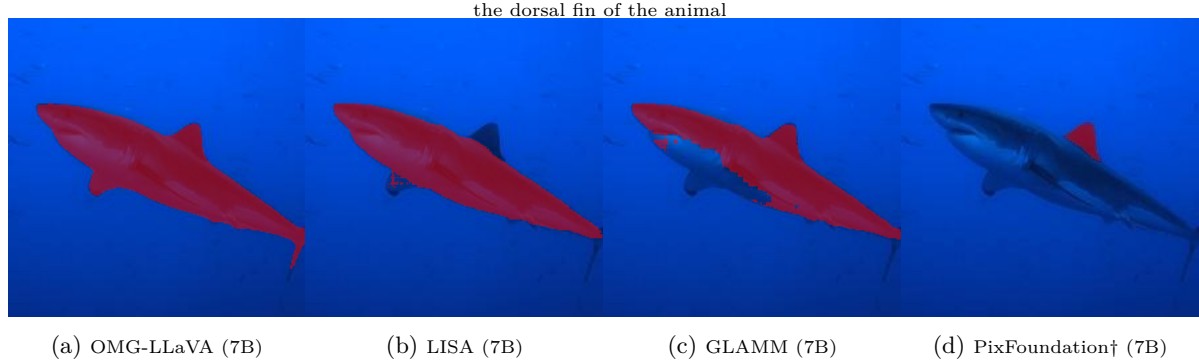

(a) OMG-LLaVA (7B)     (b) LISA (7B)     (c) GLAMM (7B)     (d) PixFoundation† (7B)

Figure 8: **PixMMVP qualitative comparison** in visual grounding following the second probing. The referred expression is on top. It shows that mining for grounding within the attention maps of standard MLLMs w/ oracle mask selection is better than MLLMs trained with mask supervision, without degrading their VQA abilities. Thus, questioning the current training recipes and design choices of pixel-level MLLMs to fully utilize the potential in their base MLLMs.

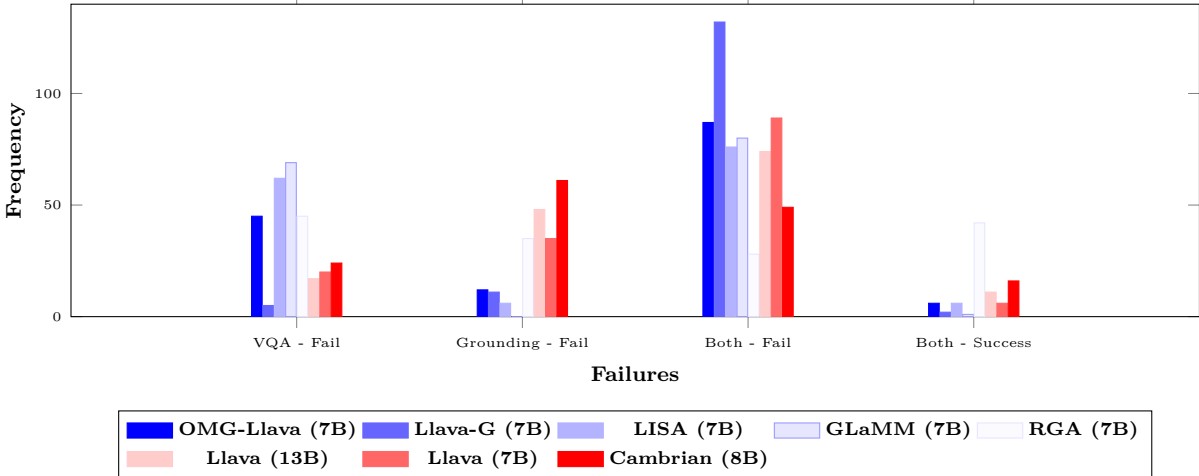

Figure 9: **PixMMVP Frequency of failures** in both visual grounding and VQA *vs.* VQA failures only *vs.* grounding only. For visual grounding, IoU $< 0.5$, is considered as a failure.

on PixMMVP using the mean intersection over union. Across both prompt variations modes, whether on the grounding prompt without the referred expression or modifying the referring expression only, RGA shows stronger results and better stability than Qwen2.5-VL. The latter exhibits higher sensitivity to the language prompt. It is mainly impacted by the prompt template, where it performs better when instructed with the "Locate" keyword than other words in the template to find the object.

Finally, our paired evaluation enables the analysis of the failures to understand whether it is stemming from grounding or VQA. Figure 9 shows the frequency of failures with respect to VQA, visual grounding or both, where the majority stem from failures in both, especially in the pixel-level MLLMs, except RGA, which shows the majority of failures are in the VQA. The standard MLLMs perform better in the VQA than grounding, as expected. However, improving pixel-level visual grounding in a naive manner that neglects the language capability eventually resulted in failures in both. On the other hand, RGA shows the highest success on both VQA and visual grounding, showing it as a promising direction, yet, as shown in the prompt sensitivity results, there are still aspects that can be improved to ensure they do not lose the language capabilities and robustness that existed in standard MLLMs. We encourage better training recipes, datasets and design choices to enable visual grounding in MLLMs without sacrificing their language abilities that are tightly coupled with grounding.

**Summary.** In summary, the state-of-the-art pixel-level MLLMs prior to the release of our benchmark have shown lower performance than simple baselines, where they specifically lag in their language abilities

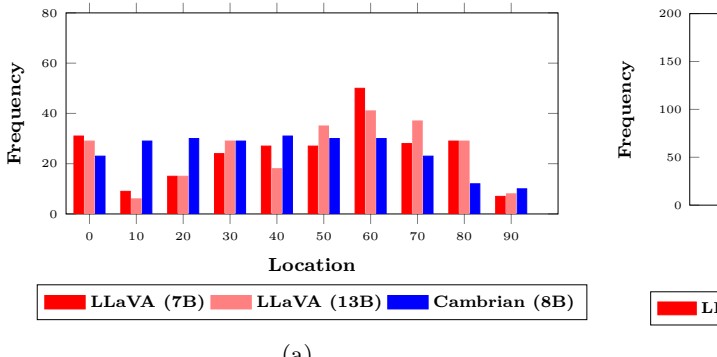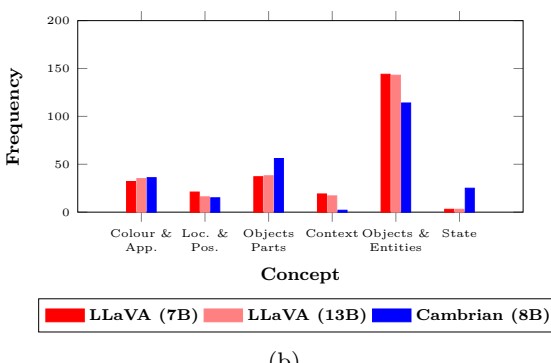

(a)                            (b)

Figure 10: **Analysis on when grounding emerges** on PixMMVP benchmark using the three base MLLMs, LLaVA 1.5 (7, 13B) and Cambrian (8B), that were not trained with pixel-level grounding supervision. We follow the second probing, then report the oracle selection. Analysis on: (a) the output location and (b) the output concept category, which coincides with the best segmentation.

and visual question answering. Prior to our work, pixel-level MLLMs were mostly ignoring the language abilities, yet we show various examples that emphasize the role of language in visual grounding. Moreover, we show that visual grounding might not coincide with the output noun phrase most similar to the referred expression, where our *oracle* upper bound and *automatic* method both surpass the attend and segment. Our interpretability tool enabled such an important insight to help understand these MLLMs better.

### 4.3 When does grounding emerge in MLLMs?

**When - location.** Taking into account the powerful performance of the *oracle* upper bound, it begs the question of when grounding emerges. We start by looking at when it emerges in terms of the location. We analyze the word/phrase location with respect to the full output text in terms of a percentage of its total length (i.e., 0% means the beginning of the text). Accordingly, Fig. 10a shows the location percentages histogram, binned at 10%, for the three base MLLMs reporting the oracle selection and evaluating on PixMMVP benchmark using the second probing. In the LLaVA 1.5 variants, the highest grounding is at the last 40%, while for Cambrian it is at the last 60%.

**When - concept.** For the second analysis, we look into the concept category that the correct output word/phrase corresponds to. The previous assumption, in other works is that grounding emerges in the exact noun/noun phrase of the object of interest. Except our analysis confirms that this is not necessarily the case. We take the correct noun/noun phrase where the grounding emerges based on the *oracle* from all three variants, then we pass it to GPT-4o to request a grouping of these concepts. It results in six main groups, which are: (i) color and appearance, (ii) location and position, (iii) object parts, (iv) context and setting, (v) objects and entities, and (vi) State. We then prompt for each of the noun/noun phrases, GPT-4o, to categorize it within these six categories. The histogram of the occurrences of these concept categories is shown in Fig. 10b. It conveys that in certain scenarios, the correct output when grounding emerges can be describing the position or the color of the object. Figure 6 shows qualitative examples of these scenarios.

**Summary.** In summary, we found that emergent pixel-level grounding might not coincide with the input referring expression. We show that grounding in MLLMs can emerge in the noun phrase that corresponds to color, position or other characteristics of the object of interest.

## 5 Conclusion

We propose two benchmarks showing that pixel-level MLLMs degrade the ability in VQA and even grounding e.g., fine-grained object parts. We propose language and visual prompt variations tailored for visual grounding. Additionally, we provide an interpretability mechanism that showed an interesting insight on when visual grounding occurs in MLLMs. Our paired benchmarks and evaluation pave the road towards better and interpretable pixel-level MLLMs.

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

## List of Appendices

## Appendix A    Additional implementation details

In this section, we cover additional details about our implementation, standard evaluation setup, prompt sensitivity evaluation setup and baselines.

**Models.** We detail the model checkpoints we use for the four pixel-level MLLMs and their variants, retrieved from HuggingFace Wolf et al. (2019) in Table S1 (left). These also include the model checkpoints used for the base MLLMs that were not trained with mask supervision. Furthermore, we provide details on the *oracle* selection mechanism. We discard the cases where the ground-truth segmentation is all background in the oracle selection in the when analysis and the point accuracy evaluation, since there is no ground-truth grounding to evaluate against. While in the quantitative and qualitative evaluation, we use an empty mask for the oracle. Similar to the automatic variant of PixFoundation, we follow their approach in filtering out the selected noun phrases for the attend and segment. We discard the ones that have a similarity less than 0.7 in the SpaCy embeddings and instead output an empty mask. This pre-processing is only needed in PixMMVP, while on PixCV-Bench we do not need such filtration across the (a+s), *automatic* or *oracle* variants. In the *oracle* upper bound, whenever the model is to be evaluated in a multiple object scenario, we take all the possible pairs of the masks and select the best pair based on the highest intersection over union.

| Model Name | Model Checkpoint | Model Name | Probing | Output Length | # Noun Phrases |
|---|---|---|---|---|---|
| LISA (7B) | xinlai/LISA-7B-v1-explanatory | LLaVA 1.5 (7B) | First | 44.2 | 2.3 |
| GLAMM (7B) | MBZUAI/GLaMM-FullScope | LLaVA 1.5 (13B) | First | 45.3 | 2.4 |
| RGA (7B) | SurplusDeficit/UniGR-7B | Cambrian (8B) | First | 313.8 | 15.2 |
| LLaVA-G (7B) | Haozhangcx/llava_grounding_gd_vp | | | | |
| LLaVA 1.5 (7B) | liuhaotian/llava-v1.5-7b | LLaVA 1.5 (7B) | Second | 92.6 | 5.2 |
| LLaVA 1.5 (13B) | liuhaotian/llava-v1.5-13b | LLaVA 1.5 (13B) | Second | 97.2 | 5.5 |
| Cambrian (8B) | nyu-visionx/cambrian-8b | Cambrian (8B) | Second | 561.3 | 27.3 |
| Qwen2.5-VL (7B) | Qwen/Qwen2.5-VL-7B-Instruct | | | | |

Table S1: **Additional details about implementation and output. Left:** Hugging Face model checkpoints used. **Right:** The average output length across PixMMVP dataset for the three base MLLMs using the first and second probing techniques.

Additionally, we provide details on the SAM model that is used in our baselines and interpretability tool, where we use the ViT-H variant. For the *automatic* and the *oracle* variants of PixFoundation with Qwen2.5-VL, we use the output point generated directly from the model instead of mining the attention maps, where we provide these results for reference. Nonetheless, we maintain the mask selection among the three masks corresponding to each point.

**Evaluation.** We also provide the details on computing the visual question answering accuracy using GPT-4o in the first probing Tong et al. (2024b) (i.e., $\mathcal{A}\dagger$), or when evaluating the language prompt sensitivity in the third probing with GPT-4o (i.e., $\mathcal{A}$ w/ GPT). We use the following prompt: *"Given the following question <QUESTION>, the correct answer is <ANSWER>. Does the following answer correctly answer the question, answer: <RESPONSE>? Respond with a Yes/No".* As for the mean intersection over union on PixMMVP (i.e., $\mathcal{M}\dagger, \mathcal{M}$), we provide two results: the results on all images, including the ones that do not have an object of interest and their referring expression is marked as "None", and when excluding this set of images. The remaining results in the appendix for the mean intersection over union exclude this set unless stated otherwise. Note that our benchmarking was conducted on an A6000 48G GPU-equipped machine.

**Output length.** We also provide additional analysis on the output length on average through PixMMVP dataset using the first and second probing schemes. Specifically, we report the output length as the number of characters in the output, and the number of noun phrases extracted from it. The reason to show this, since it has a relation to the number of noun phrases and consequently the number of masks our interpretability mechanism is selecting among. Table S1 (right) shows the average output length computed across PixMMVP dataset, comparing the three base MLLMs. We notice that Cambrian (8B) generates longer outputs with a considerable margin than LLaVA variants. Hence, we believe the superiority of the *oracle* upper bound with Cambrian in the grounding has strong correlation to producing longer outputs with more attention maps to mine and select from, than LLaVA variants. Nonetheless, it makes it more challenging for the automatic baseline.

## Appendix B   Additional quantitative analysis

### B.1   PixFoundation using open-source models

In our *automatic* baseline, we replace GPT-5.1, which is a closed-source model, with another open-source model, in our case, Cambrian (8B). Table S2 shows the results on PixMMVP for PixFoundation, the *automatic* baseline that still surpasses the best pixel-level MLLM, OMG-LLaVA, that was before the release of our benchmark. More importantly, this baseline confirms that even with the use of a self-contained model such as Cambrian, without additional help from GPT-5.1, it can still compete with these unified pixel-level supervised models.

### B.2   Conventional visual grounding benchmarks

We evaluate our upper bound on the conventional benchmarks for Ref-COCO/RefCOCO+/RefCOCOg Kazemzadeh et al. (2014); Mao et al. (2016) with respect to state-of-the-art

| Method | PixMMVP | | | | |
|---|---|---|---|---|---|
| | $\mathcal{A}$† | $\mathcal{A}$ | $\mathcal{M}$† | $\mathcal{M}$ | $\mathcal{S}$ |
| OMG LLaVA (7B) ◇ | 12.0 | 12.0 | 13.8 | 38.8 | 18.3 |
| LLaVA 1.5 (7B) + (a+s) | 27.3 | 28.0 | 7.8 | 14.1 | 18.8 |
| LLaVA 1.5 (13B) + (a+s) | 39.3 | 30 | 8.5 | 13.9 | 20.5 |
| Cambrian (8B) ∗ + (a+s) | **52.0** | **52.0** | 15.5 | 13.3 | 23.9 |
| Cambrian (8B) ∗ + PixFoundation (Ours) | **52.0** | **52.0** | 44.7 | 41.4 | 48.1 |
| Cambrian (8B) ∗ + PixFoundation⋆ (Ours) | **52.0** | **52.0** | 24.6 | 25.8 | 34.5 |

Table S2: **Open-source automatic mask selection.** Comparison of pixel-level MLLMs to our automatic baseline that relies on Cambrian (8B), an open-source model, for the automatic selection (PixFoundation⋆) on PixMMVP. Instead of using GPT-5.1 which is closed source (PixFoundation). Best results are bolded.

| Method | refCOCO | | | refCOCO+ | | | refCOCOg | |
|---|---|---|---|---|---|---|---|---|
| | val | testA | testB | val | testA | testB | val | test |
| LISA | 74.1 | 76.5 | 71.1 | 62.4 | 67.4 | 56.5 | 66.4 | 68.5 |
| LISA (ft) | 74.9 | 79.1 | 72.3 | 65.1 | 70.8 | 58.1 | 67.9 | 70.6 |
| OMG LLaVA (7B) ◇ | 75.6 | 77.7 | 71.2 | 65.6 | 69.7 | 58.9 | 70.7 | 70.2 |
| OMG LLaVA (7B) ◇ (ft) | **78.0** | 80.3 | **74.1** | 69.1 | 73.1 | 63.0 | 72.9 | 72.9 |
| Cambrian (8B) ∗ + PixFoundation† | 77.8 | **81.1** | 72.8 | **73.2** | **79.5** | **70.1** | **75.5** | **77.5** |

Table S3: **RefCOCO benchmark** comparison of pixel-level MLLMs to one of our provided baselines and our upper bound using cIoU. Best results are bolded.

methods in pixel-level MLLMs prior to the release of our benchmark. Table S3 shows that our upper bound that relies on extracting the output noun phrases, mining their attention maps and selecting the best mask output in relation to these attention maps is competitive to state-of-the-art methods. However, the main goal of PixFoundation is to inspect when visual grounding occurs in MLLMs with respect to the output token. We still show the results on refCOCO benchmarks for reference. Note that unlike refCOCO, our benchmarks pose a greater challenge for pixel-level MLLMs since they are focused on vision-centric tasks that require grounding to answer the questions. Furthermore, our paired evaluation enables the evaluation of the grounding in relation to the VQA task to better study these MLLMs. Finally, unlike refCOCO variants, our PixMMVP especially provides an out-of-distribution challenging evaluation benchmark, where the majority of these MLLMs have been trained on refCOCO training data.

### B.3 When grounding emerges - PixCV-Bench

In Fig. S1a, we show the analysis of when grounding emerges on PixCV-Bench in terms of the location in the output. It is worth noting that PixMMVP is more challenging than PixCV-Bench, evidently from the reported IoU and accuracy metrics on both with respect to Table 2 and Table 3. It seems on the less challenging dataset PixCV-Bench, grounding tends to emerge frequently near the beginning of the output. This might relate to PixMMVP being more challenging in terms of the level of reasoning than PixCV-Bench or the fact that PixMMVP poses a harder referring segmentation task than PixCV-Bench, which mostly uses the class names. Another difference is that PixMMVP is out of the distribution of the seen datasets for most of these MLLMs. However, the consistent finding among both datasets is that grounding can emerge coinciding with various concept categories, whether location, color or state, as shown in Fig. S1b. Note that across this analysis, we compute the frequency per object in the referred expression corresponding to the visual question. Hence, if we have two objects in one visual question, such as in the relative positioning questions, each object's concept, corresponding to the emergence, is computed as part of our analysis.

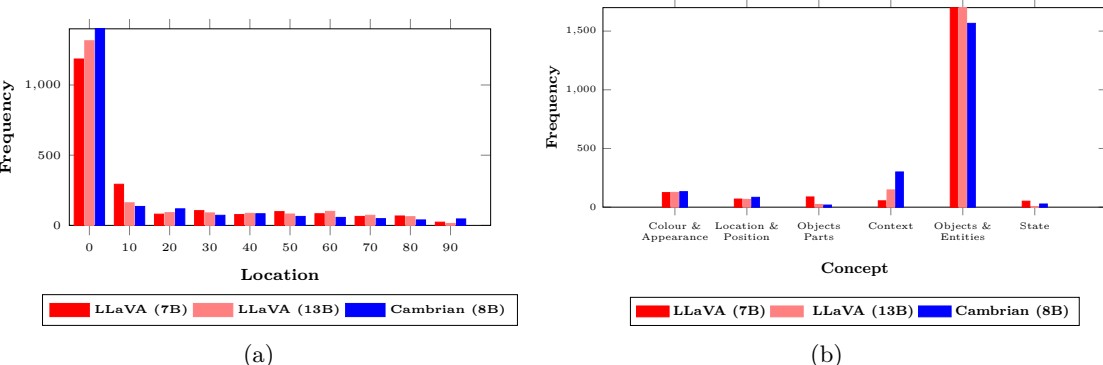

Figure S1: **Analysis on when grounding emerges on PixCV-Bench** benchmark using the three base MLLMs, LLaVA 1.5 (7, 13B) and Cambrian (8B), that were not trained with pixel-level grounding supervision. We follow the second probing then report the oracle selection. Analysis on: (a) the output location and (b) the output concept category, which coincides with the best segmentation.

### B.4 Pixel-level MLLMs step-wise ablation

We perform an ablation on two sets of models in Table S4, comparing the inference with vs. without the segmentation decoder, and with vs. without mask supervision during training. This helps provide a more fine-grained analysis of potential directions to improve pixel-level MLLMs. We evaluate across the standard separate prompts per task, requesting VQA or visual grounding, and then we evaluate the two paired tasks following the previous evaluation protocols. We also use an additional probing here, which prompts MLLMs with one prompt for the joint tasks, requesting the model to do both. We use the prompt *"<Question>? <Choices>. Answer this question while identifying the object you need to answer it (with grounding)."* We report the VQA accuracy metrics, $\mathcal{A}\dagger, \mathcal{A}$, where the former can evaluate the models' ability to generate free-form text from the multiple choices provided and the latter evaluates the models' ability to follow instructions to generate one option letter. We also report the visual grounding metrics as point-accuracy, $\mathcal{PA}$, indicating whether it falls in the groundtruth mask or not. Moreover, we report the normalized distance, $\mathcal{D}_{\mathrm{norm}}$, between the predicted point grounding with respect to the center of the groundtruth mask, where we normalize by the diagonal length of the image. Note that in rare scenarios, a gain in point accuracy does not necessarily reflect a reduction in the normalized distance. This is due to scenarios where predicted points lie near the mask boundaries but are not accurate enough to coincide with the groundtruth mask. For each set, the model (RGA/LISA) is compared to its backbone MLLM (Qwen2.5-VL/LLaVA-1) without mask supervision. Similarly, for each model, we evaluate its full inference with an output mask vs. its partial model without the segmentation decoder, analyzing the point grounding similar to PixFoundation†. Specifically, we select the closest predicted point from all the output tokens to the center of the groundtruth mask. When the model outputs a mask, we use its center as the predicted point.

Across both probing with separate or joint tasks prompts, both LISA and RGA still suffer in free-form text generation, as shown in $\mathcal{A}\dagger$, unlike their backbone MLLM, which was not trained with mask supervision. Nonetheless, RGA is better at the VQA accuracy when evaluating its ability to generate an instruction letter in the separate prompts per task with respect to LISA. Both completely fail in the joint tasks prompt probing. This is because of the way the model was heavily instruction-tuned to respond to multiple-choice questions in relation to a specific set of prompt formats. Yet RGA is a step in the right direction to ensure the model is instruction-tuned for both VQA and visual grounding with proper mixing of both tasks, unlike LISA. The backbone MLLM across both sets show persistently that it can still respond to the joint tasks prompt, instructing the model to answer the question and ground the object of interest simultaneously. Hence, these show a better language understanding than their counterparts trained with mask supervision.

When looking at the visual grounding ability, the full models outperform or are close to the partial models and the base MLLM across both separate and joint tasks prompt settings. The use of the segmentation decoder, in the full models, improves the normalized distance and makes grounding more accurate than the partial

| | Method | Training w/ Mask Supervision | Inference w/ Seg. Decoder | Seperate Tasks Prompts $\mathcal{A}\dagger, \mathcal{A}$ | $\mathcal{PA}\uparrow, \mathcal{D}_{norm}\downarrow$ | Joint Tasks Prompt $\mathcal{A}\dagger, \mathcal{A}$ | $\mathcal{PA}\uparrow, \mathcal{D}_{norm}\downarrow$ |
|---|---|---|---|---|---|---|---|
| Set A | RGA- Full | ✓ | ✓ | 5.3, **51.3** | **72.1**, 0.08 | 1.3, - | 56.3, **0.17** |
| | RGA - Part | ✓ | ✗ | 5.3, **51.3** | **72.1**, 0.18 | 1.3, - | **57.5**, 0.21 |
| | Qwen2.5-VL | ✗ | ✗ | **54.7**, 38.0 | 57.9, 0.22 | **32.7**, - | 42.5, 0.23 |
| Set B | LISA - Full | ✓ | ✓ | 7.3, - | 57.1, 0.12 | -, - | 50.6, **0.17** |
| | LISA - Part | ✓ | ✗ | 7.3, - | 51.4, 0.24 | -, - | 48.6, 0.26 |
| | LLaVA-1 | ✗ | ✗ | 14.7, - | 56.3, 0.24 | 12.0,- | 51.4, 0.28 |

Table S4: **Step-wise ablation of pixel-level MLLMs** with vs. without mask supervision during the MLLM training, and with vs. without the segmentation decoder used during inference. We report both the VQA and visual grounding metrics for point accuracy and normalized distance. We evaluate under our proposed three probing techniques for the separate tasks prompts. We also evaluate with a fourth probing that prompts the model to generate both the question's answer and grounding, under the joint tasks prompt. Best results are bolded.

models, which select among points corresponding to the output tokens. The partial models outperform the backbone MLLM in the normalized distance, without mask supervision or a segmentation decoder. Hence, training with mask supervision does improve the emergent visual grounding ability of these models, learned within their attention maps. However, considering the previous findings in the VQA, especially in the joint tasks prompt, even the best model, RGA, still loses some of its backbone language understanding. We believe these findings do not discard the possibility of exploring novel pixel-level MLLMs with output masks on the language side through generating polygonal points, yet this is out of the scope of our work. Additional studies focused on instruction tuning, training recipes and pixel-level MLLMs architectures inspired by our findings are left for future work.

### B.5   Run-time component-wise ablation

In Table S5 we report component-wise run-time ablation in seconds, in addition to our previously reported accuracy and robustness scores on PixMMVP dataset. We use LLaVA 1.5 (7B) in our PixFoundation mechanism and the baselines. The simple baseline, combining non-unified components already existent in the unified ones, is better in terms of accuracy, robustness scores and on-par in run-time, vs. the unified models e.g., LISA. Similarly, for the simple baseline vs. the (a+s) baseline. This clearly motivates our benchmarking effort. Our PixFoundation maintains the best accuracy and robustness scores, surpassing the simple baseline, yet it entails a reasonable computational overhead. Note that PixFoundation is mainly meant as an interpretability mechanism, which reduces the need for real-time performance and is rarely considered for interpretability techniques. Nonetheless, considering PixFoundation improvement over the (a+s) baseline in the scores, it shows the significance of our interpretability insights, since our insights on the emergent grounding with respect to the output tokens directly relate to this performance boost. All experiments are reported on an A6000 GPU, the GPT-5.1 run-time relies heavily on the internet speed and bandwidth, where we report run-time at 100 mb/s average internet speed approximated from 15 mb/s.

## Appendix C   Additional qualitative analysis

### C.1   Studying the OOD characteristic of PixMMVP & PixCVBench

In order to study how the new benchmarks form a more challenging setting for MLLMs trained with RefCOCO dataset, we provide a TSNE Van der Maaten & Hinton (2008) qualitative analysis. First, we extract embeddings from CLIP that were used across multiple MLLMs, including LLaVA, for the following RefCOCO variants (RefCOCO val, RefCOCO testA, RefCOCO+ val, RefCOCO+ testA, RefCOCOg val, RefCOCOg test). We also use our benchmark splits (PixMMVP, PixCVBench (ADE20K), PixCVBench (COCO)). We provide a relative TSNE plot showing source, reference and target datasets for each. The source and reference are in-distribution, both stemming from RefCOCO, while the target is the one to study, which is selected from our three splits. Figure S2 across all the RefCOCO splits with respect to our splits,

| Method | $\mathcal{S}$ | $\mathcal{S}_{\text{sensitivity}}$ | Run-time (s) Component-wise | Total |
|---|---|---|---|---|
| PixFoundation - Point & Noun Phrase Ext. PixFoundation - SAM PixFoundation (GPT-5.1)- Filtration PixFoundation (GPT-5.1) - Mask Selection | **33.6** | **25.0** | 1.0 0.5 0.2 1.2 | 2.9 |
| (a+s) - Point & Noun Phrase Ext. (a+s) - SAM (a+s) - Spacy Similarity | 18.8 | 17.5 | 1.0 0.5 0.02 | 1.52 |
| LISA | 12.5 | - | 1.4 | **1.4** |
| Our Baseline - Box Grounding Our Baseline - SAM | 33.0 | 20.2 | 0.9 0.5 | **1.4** |

Table S5: **Runtime ablation** for the components used in the simple baseline, (a+s) and our PixFoundation vs. LISA one of the unified pixel-level MLLMs. The accuracy and prompt variations robustness scores, $S, S_{\text{sensitivity}}$, are reported on PixMMVP in standard and prompt sensitivity settings. It shows our non-unified simple baseline surpassing all other models in terms of scores and run-time, except for our PixFoundation scores, motivating our benchmarking effort. It also shows our PixFoundation, which is mainly meant as interpretability mechnism, as the best in terms of the scores, yet entails a reasonable computational overhead. Best and second best results are bolded and underlined, resp.

PixCVBench (COCO), as expected, is more in-distribution with respect to both the source and reference datasets, unlike PixMMVP and PixCVBench (ADE20K). The latter two splits show clusters that are less interconnected with these. This is simply because both stem from other datasets than COCO, and as such, they are more out-of-distribution with respect to our COCO split.

## C.2 Baselines ablation

We show the qualitative ablation between the baseline and our interpretability mechanism using the best base MLLM Cambrian (8B) in Fig. S3 on PixMMVP. The three confirm that there is pixel-level grounding emerging in MLLMs that were not trained with mask supervision. Nonetheless, it shows that identifying when that grounding emerges is equally important in retrieving the best segmentation of the referring expression. The first baseline, attend and segment, assumes the alignment between the attention map that can be mined for the segmentation mask and the noun phrase that has the highest correspondence to the queried category or noun phrase. Our findings quantitatively and qualitatively show otherwise, where grounding can emerge in different output tokens. It also shows the *oracle* upper bound for mask selection, PixFoundation†, exhibiting better segmentation than the attend and segment, confirming the aforementioned finding. Additionally, it shows that our simple automatic mechanism, PixFoundation, surpasses the attend and segment as well on PixMMVP.

## C.3 Attention maps visualization

In this section, we visualize the normalized attention maps, $\tilde{A}$, in Fig. S4. We show two examples for Cambrian (8B) from PixMMVP using the first probing where we directly prompt the model with question and options. The first row shows outstanding ability to visually ground the different noun phrases from the output text. The full output text of the first row example is: *"The image provided is a cake designed to resemble a minion from the D̈espicable Mef̈ranchise. It is not a living creature and therefore cannot smile or have a tongue out. The cake is an inanimate object, crafted to mimic the appearance of a minion, which is a fictional character from the animated movie series. The design elements such as the **yellow skin, blue overalls, and goggles** are characteristic of the minions' appearance in the films."* The visualization shows how the maximally attended locations for the last three noun phrases correspond to the correct locations in the image.

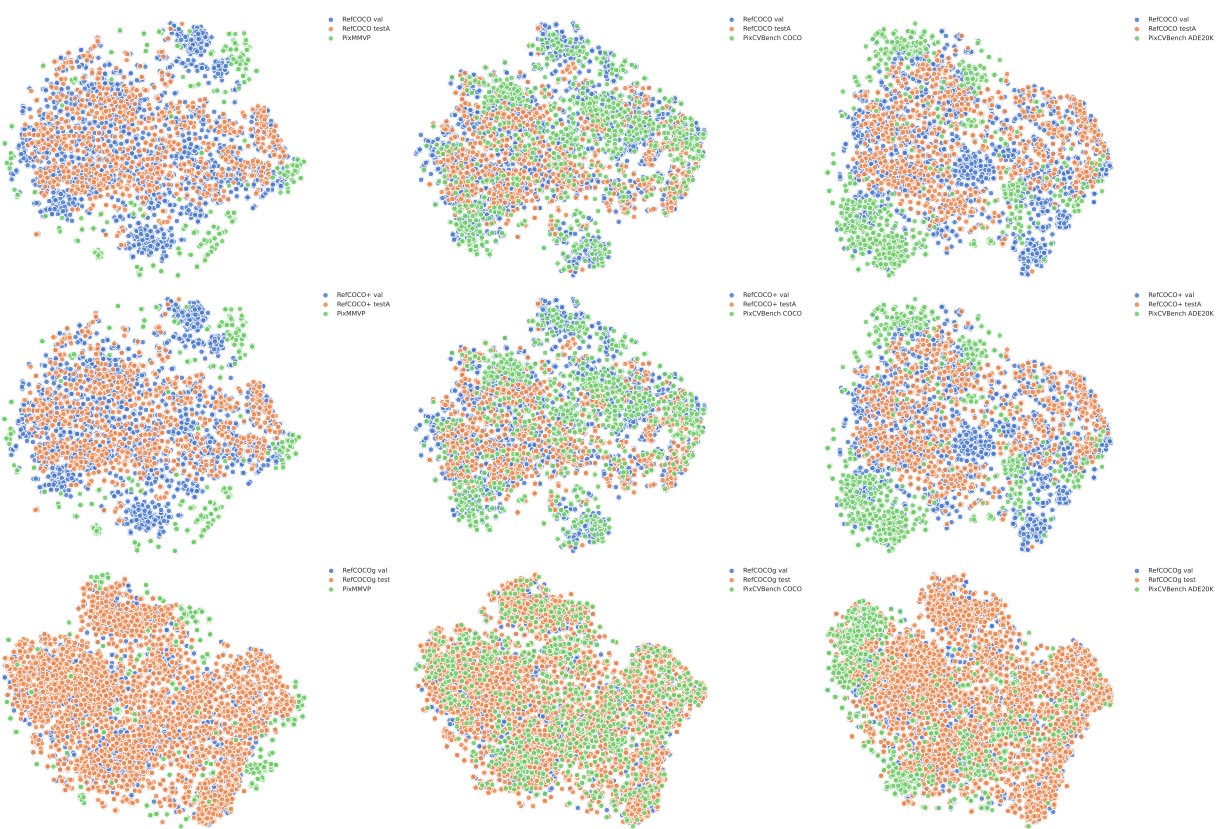

Figure S2: **Relative TSNE** to show the out-of-distribution nature of PixMMVP and PixCV-Bench (ADE20K) part with respect to RefCOCO variants, where the latter dataset is used in standard training & evaluation of MLLMs equipped with visual grounding.

The second output text corresponding to the example shown is; *"The butterfly's feet, also known as tarsi, are not distinctly visible in this image due to the angle and the butterfly's wings being open. However, we can infer their presence and approximate location. Monarch butterflies have* **six legs***, with the hind legs being the longest and the* **front legs** *being the shortest. The legs are typically slender and have a* **hairy texture***, which aids in gripping onto surfaces. In this image, the legs are likely located at the bottom of the butterfly's body, just below the abdomen, and are probably in contact with the leaf it is perched on."* The attention maps highlight what we suspect is a failure where the MLLM mistakes the antenna of the butterfly for front legs. Such hidden failures that do not necessarily affect the correctness of the answer, are still important to study and we believe our tool with the *oracle* upper bound can be used to inspect this further. Finally, we find that these attention maps in both examples are not sufficiently accurate to be used for segmentation directly, yet when paired with a powerful segmentation method like SAM it provides a good segmentation performance.

### C.4    When does grounding emerge?

We show additional examples of when grounding emerges in multi-modal large language models, specifically in the LLaVA 1.5 (7B) variant, using the second probing to prompt the model to segment what is in the referring expression. Figures S5, S7, S6 and S8 show the corresponding predicted masks for the grounding that emerged, highlighted in red with the maximum attention point as a black circle. Figure 6 shows the aforementioned four examples with the referred expression, the concept category and the noun phrase corresponding to the best grounding using the *oracle* selection and the full output text. It clearly shows that the correct output token can correspond to location or color, but not necessarily the queried referring expression. While some of the noun phrases and their masks, from the SAM point prompting, correspond

Attend and Segment Cao et al. (2024) (Cambrian (8B))

PixFoundation (Cambrian (8B))

PixFoundation† (Cambrian (8B))

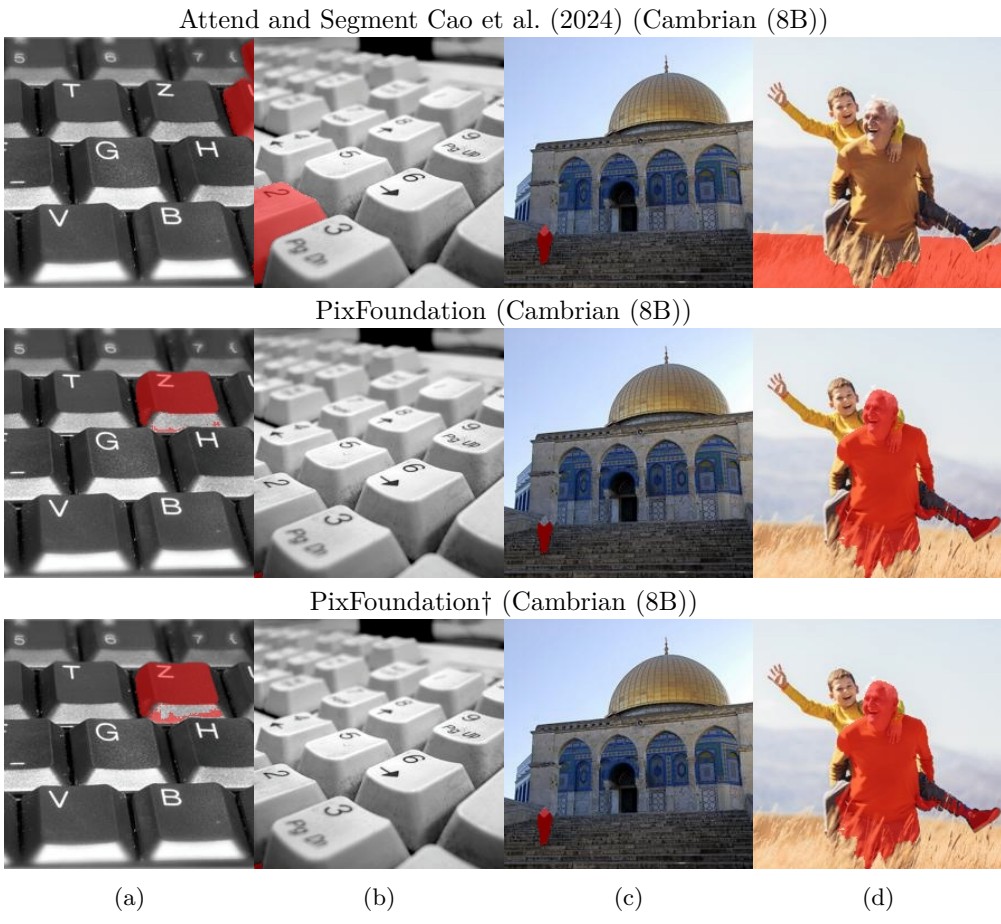

(a)        (b)        (c)        (d)

Figure S3: **Qualitative ablation** for the baseline and our interpretability tool using the base MLLM, Cambrian (8B), ablating the different schemes for mask selection. We use the second probing to prompt the MLLM to identify the referred expression. The referring expressions for these examples are as follows: (a) the key "z", (b) the key "z", (c) people, (d) the elderly person. Predictions are highlighted in red.

to what the noun phrase is describing. It is not always the case, for example, in Fig. S6 "the flame" was not able to highlight the correct object, yet it appeared in the noun phrase corresponding to the location "the top". While few scenarios might have the grounding coinciding with multiple noun phrases, such as in Fig. S5, "a butterfly" and "orange wings". Nonetheless, it is still an important insight that the segmentation can emerge corresponding to noun phrases that do not correspond to the exact referred expression. Our PixFoundation† serves as an interesting tool to interpret and understand how MLLMs work and reason to produce the final output with the *oracle* selection as an upper bound.

In summary, we provide four evidences that pixel-level grounding can emerge corresponding to noun phrases that do not match the exact referred expression, as follows: (i) The attend and segment that rely on SpaCy embeddings lag behind our *automatic* and *oracle* mask selection, indicating that the noun phrases closest to the referred expressions are not necessarily where the optimal segmentation emerges. (ii) The point accuracy evaluation has confirmed this further, which removes any confounding factors from using three masks per noun phrase in the mask selection due to the point prompt ambiguity. The point accuracy only evaluates the emergent grounding with respect to the noun phrases. (iii) We show quantitative analysis on the location and the concept categories of the noun phrases where the grounding emerges, which confirms the previous result. (iv) We show qualitative analysis to confirm this further.

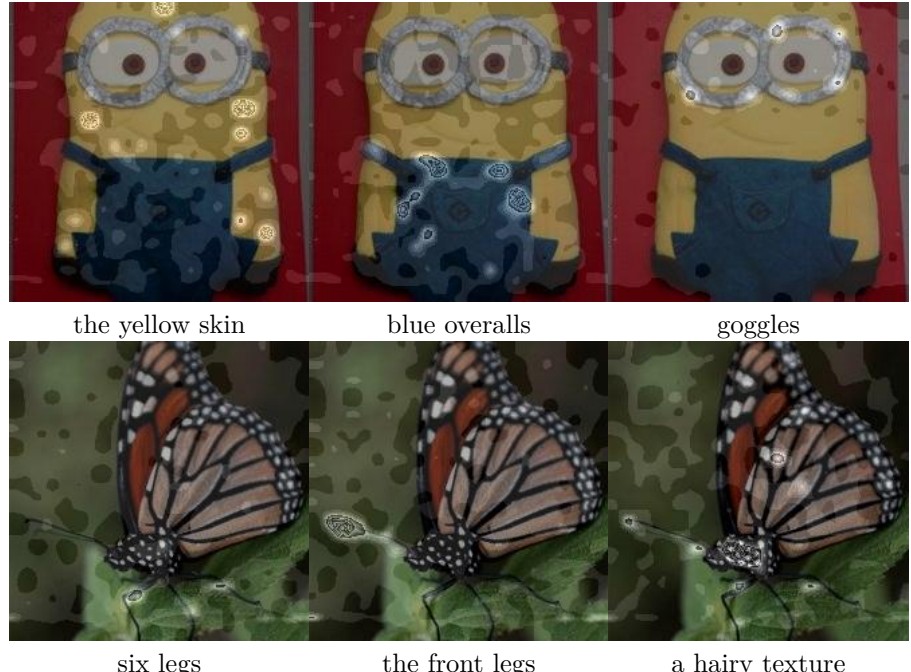

Figure S4: **Normalized attention maps visualization** showing the noun phrase and its corresponding attention in the output text for two PixMMVP examples using Cambrian (8B) base MLLM. While the attention maps can not be directly used as segmentation, yet it provides initial locations for the maximally attended pixels corresponding to what the model is looking at. In certain scenarios it exactly aligns with the noun phrase describing it as in the two examples. Yet in certain scenarios as we showed earlier, the grounding of the referred expression in question emerges with other noun phrases describing it.

### C.5 PixMMVP additional qualitative results

Figure S9 shows additional results on PixMMVP benchmark comparing different pixel-level MLLMs with our *oracle* baseline using LLaVA 1.5 (7B). While GLAMM shows strong pixel-level visual grounding yet we have shown earlier that it is almost incapable of visual question answering, which renders the model weak for general-purpose tasks.

On the other hand, OMG-LLaVA shows a better balance in pixel-level visual grounding and visual question answering as previously detailed. Nonetheless, the simple mining of attention maps from LLaVA 1.5 (7B) using the *oracle* selection which we call PixFoundation† shows the strongest capability in both grounding and VQA. In fact, certain MLLMs that were trained with pixel-level visual grounding, such as LISA, have degraded the performance with respect to the hidden information already existing in powerful MLLMs that were not trained with such supervision.

### C.6 PixCV-Bench additional qualitative results

Figure S10 shows qualitative results on PixCV-Bench. It shows that pixel-level MLLMs struggle with segmenting the object annotated by the red box, unlike our *oracle* baseline, PixFoundation†. Indeed the attention maps from these MLLMs are looking at the right object annotated by the red box without receiving any pixel-level grounding supervision during training.

## Appendix D    Failure Cases Analysis

In this section, we conduct an additional failure case analysis of pixel-level MLLMs and our baselines qualitatively and quantitatively.

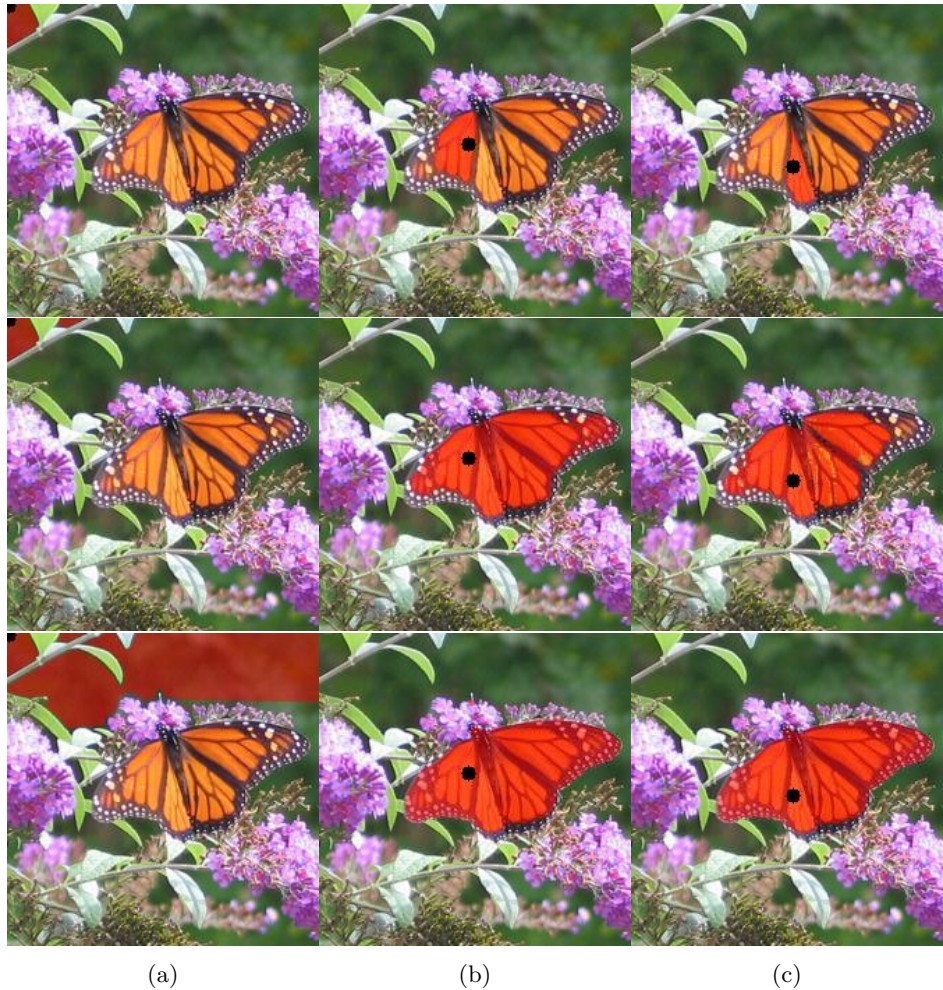

(a)                  (b)                  (c)

Figure S5: **First example of when grounding emerges**, corresponding to Image 1 in Fig. 6 main. Each column corresponds to a noun phrase and shows three potential SAM predicted masks highlighted in red using the maximum attention point of this noun phrase as a point prompt, highlighted as a black circle. It shows the output from mining the attention maps for pixel-level grounding using LLaVA 1.5 (7B) base MLLM. Noun phrases: (a) the image, (b) a butterfly, (c) orange wings.

## D.1 Failures in visual question answering

We start with a fine-grained quantitative analysis of how the studied models perform across PixMMVP and PixCV-Bench. For PixMMVP we follow their scheme to identify the nine visual patterns and report the model's accuracy with each pattern in Fig. S11. Similarly, we show fine-grained analysis relying on the tasks for the two datasets (ADE20K and COCO) in Fig. S12.

PixMMVP results show that the majority of pixel-level MLLMs, highlighted in blue, suffer in the state, orientation and quantity related tasks. On the other hand, relational context, color and presence of features show the best performance with pixel-level MLLMs. Nonetheless, across all the visual patterns, the MLLMs that were not trained with pixel-level supervision persistently surpass these pixel-level MLLMs with a considerable margin. PixCV-Bench, similarly shows the count task is more challenging than the relational positioning. It also shows that ADE20K dataset serves as a more challenging dataset than COCO.

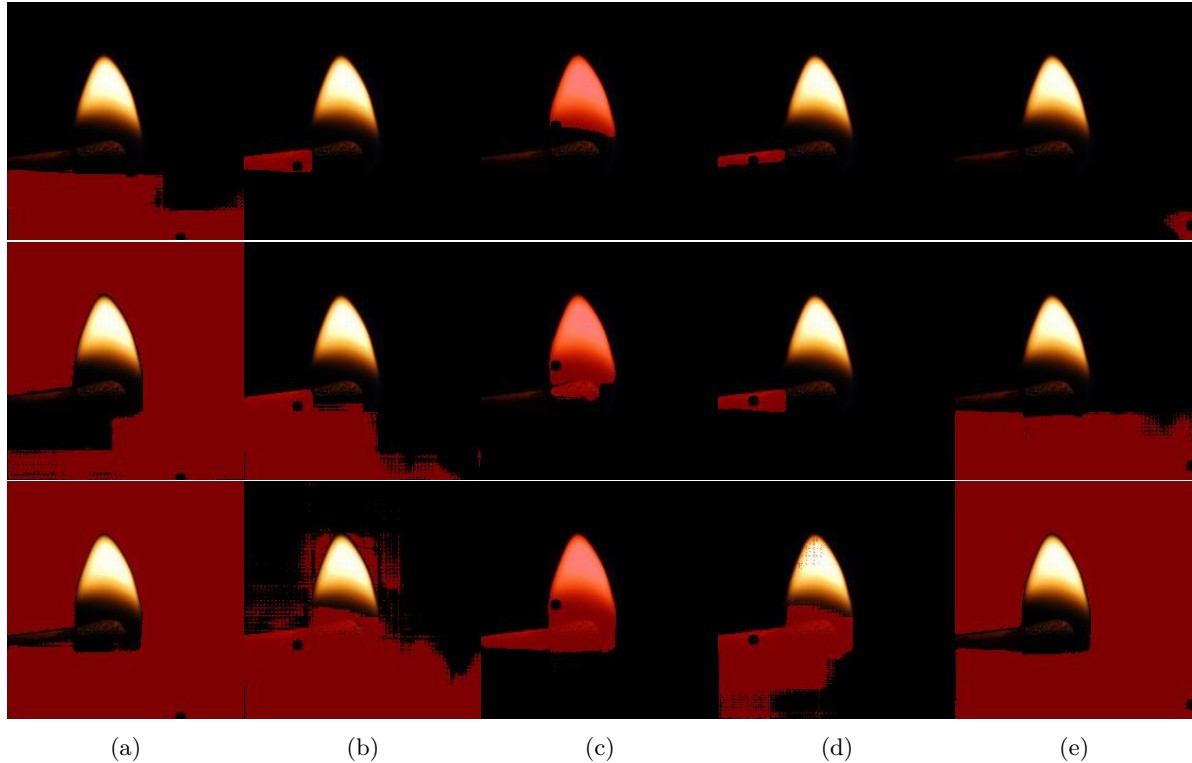

(a)        (b)        (c)        (d)        (e)

Figure S6: **Second example of when grounding emerges,** corresponding to Image 3 in Fig. 6 main. Each column corresponds to a noun phrase and shows three potential SAM predicted masks highlighted in red using the maximum attention point of this noun phrase as a point prompt, highlighted as a black circle. It shows the output from mining the attention maps for pixel-level grounding using LLaVA 1.5 (7B) base MLLM. Noun phrases: (a) The flame, (b) the match, (c) the top, (d) the image, (e) darkness

### D.2  Failures in pixel-level visual grounding

Finally, we show qualitatively the failure cases of the *oracle* upper bound in Fig. S13. It shows failures in segmenting all the object instances in the first row, since the current point prompting assumes one connected component corresponding to each expression. However, certain scenarios, such as the image with the spots on the animal, can lead to these failures in the oracle even when the localisation of some of these is correct. Mechanisms that solve this multi instance scenarios of the same object are left for future work. Potential strategies would study the top peaks in the attention maps with clustering techniques to identify these multiple instances that can be used within our PixFoundation mechanism.

Another failure occurring such as in the second row stems from ambiguity in the referring expression itself or failures from SAM identifying the separation between the wall and the ceiling. Hence, the oracle upper bound is generally inheriting SAM failures. However, its main purpose of showing that the hidden information within powerful MLLMs is sufficient to perform pixel-level grounding is achieved, and even surpasses some of the pixel-level MLLMs without degrading their VQA abilities.

## Appendix E  Impact statement

Multi-modal large language models are widely used in various applications, such as robotics, medical image processing and remote sensing. The pixel-level understanding within such MLLMs is necessary for such applications that require the localization and even in certain scenarios the delineation of the boundaries for the objects of interest. It is even more important to maintain a good chat performance and visual

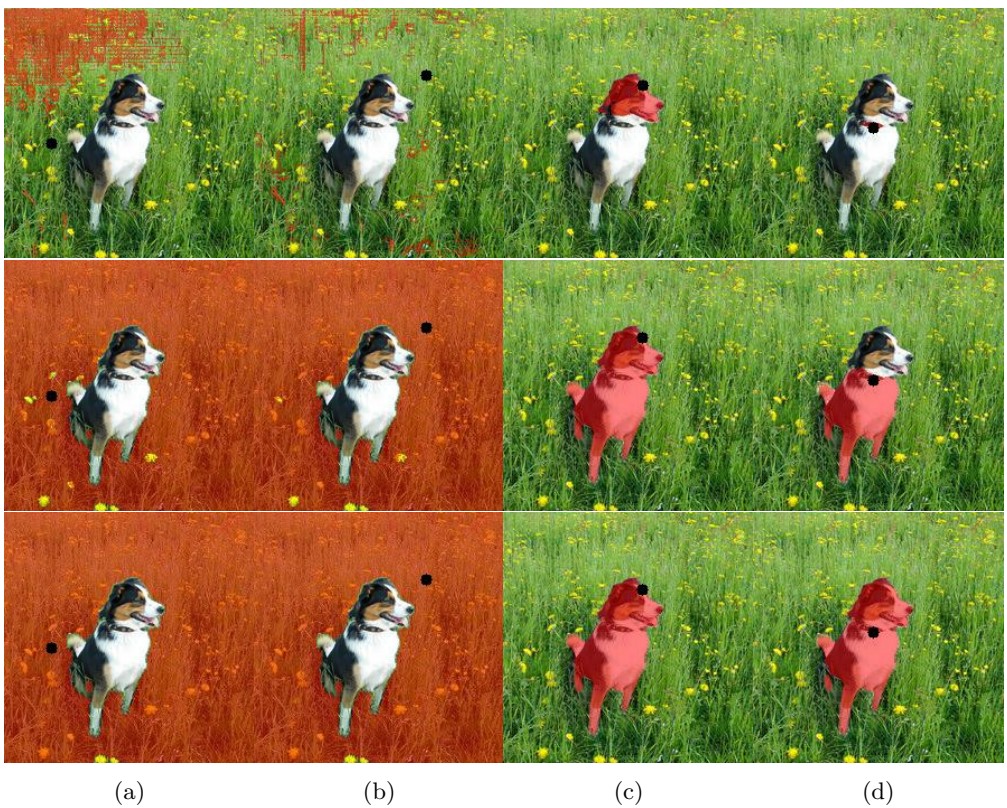

|  |  |  |  |
|:---:|:---:|:---:|:---:|
| (a) | (b) | (c) | (d) |

Figure S7: **Third example of when grounding emerges,** corresponding to Image 6 in Fig. 6 main. Each column corresponds to a noun phrase and shows three potential SAM predicted masks highlighted in red using the maximum attention point of this noun phrase as a point prompt, highlighted as a black circle. It shows the output from mining the attention maps for pixel-level grounding using LLaVA 1.5 (7B) base MLLM. Noun phrases:(a) the dog's face, (b) the scene, (c) a black and white dog, (d) a black nose.

question answering ability in such applications as well. In our work, we have investigated the shortcomings of pixel-level MLLMs while providing more challenging benchmarks for these, to improve them further.

However, as with many other AI advancements there are risks that could be entailed from the deployment of such models. There could be inherent biases emerging in such pixel-level MLLMs impacting various under-represented groups. We think that our benchmarking efforts and providing a tool to understand the pitfalls in the understanding and reasoning of these models could be an initial direction for mitigating such biases. Nonetheless, we leave it for future work to explore this further.

## Appendix F    Limitations

Note that our interpretability mechanism does entail a computational overhead with the use of the mask selection process. Nonetheless, the benefit from exploring what is already learned in these MLLMs through mining the attention maps with an understanding of when grounding emerges, provides greater benefit to interpretability. We believe interpretability of MLLMs is a crucial aspect when following a responsible approach to AI.

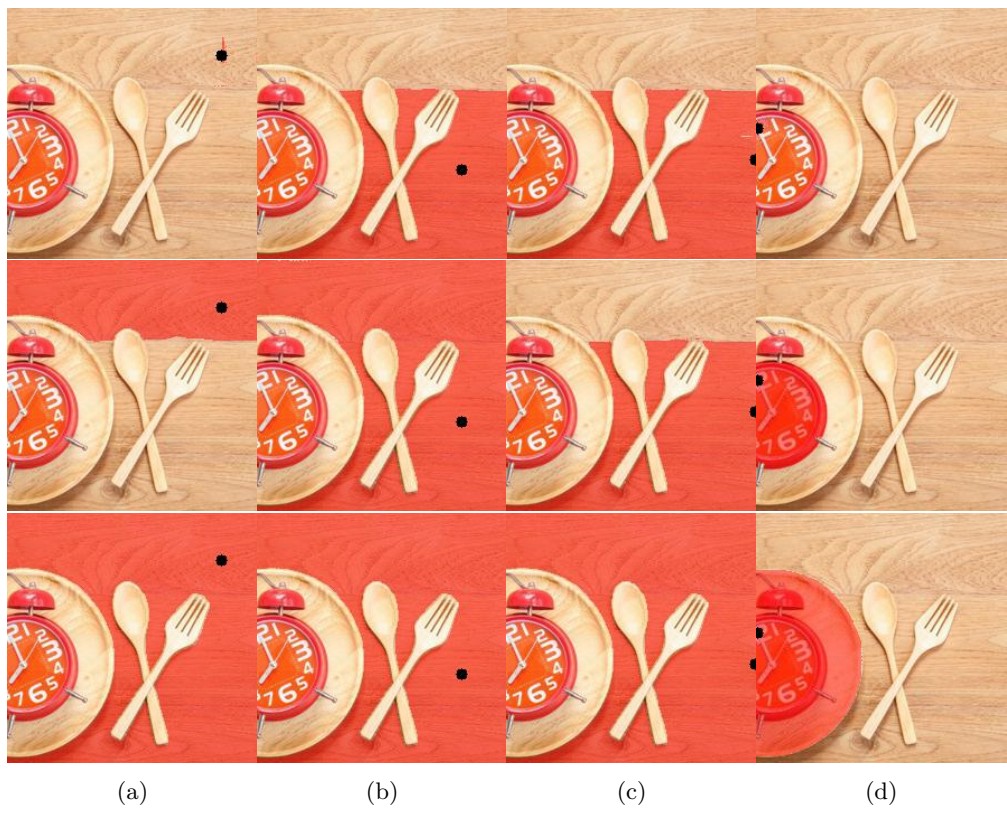

(a)        (b)        (c)        (d)

Figure S8: **Fourth example of when grounding emerges,** corresponding to Image 161 in Fig. 6 main. Each column corresponds to a noun phrase and shows three potential SAM predicted masks highlighted in red using the maximum attention point of this noun phrase as a point prompt, highlighted as a black circle. It shows the output from mining the attention maps for pixel-level grounding using LLaVA 1.5 (7B) base MLLM. Noun phrases: (a) The minute hand, (b) the clock, (c) the scene, (d) the 12 o'clock position.

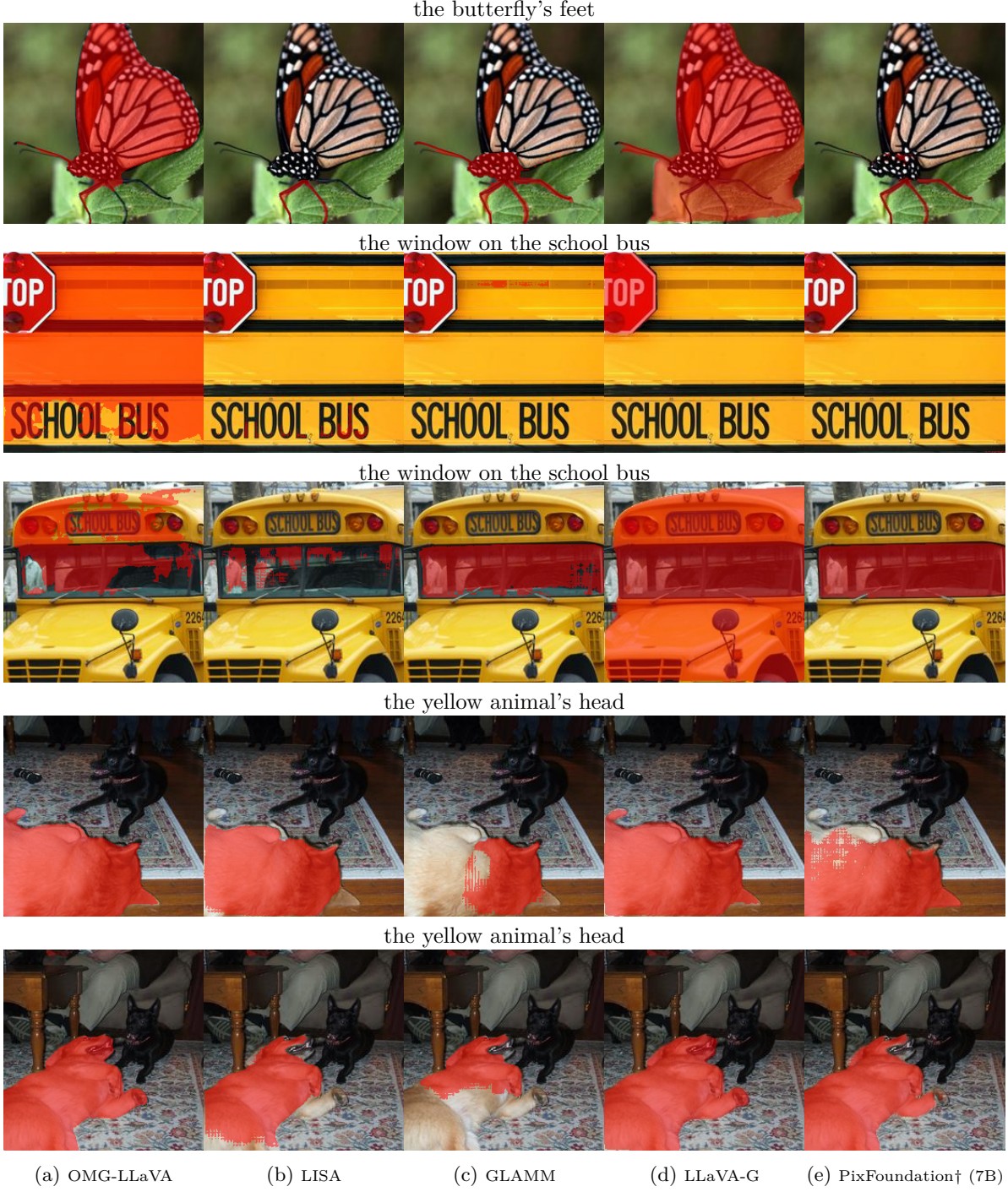

the butterfly's feet

the window on the school bus

the window on the school bus

the yellow animal's head

the yellow animal's head

(a) OMG-LLaVA (b) LISA (c) GLAMM (d) LLaVA-G (e) PixFoundation† (7B)

Figure S9: **PixMMVP qualitative comparison** between the pixel-level visual grounding following the second probing. The referred expression used in the segmentation is shown on top of each row. It shows persistently that mining for the grounding within attention maps of MLLMs that were not trained with pixel-level grounding supervision and using the oracle selection outperforms the pixel-level MLLMs. It clearly shows the oracle excels in identifying fine-grained object parts and descriptions that other pixel-level MLLMs are not necessarily capable of. The second best performance is GLAMM, yet we showed it is completely incapable of performing visual question answering.

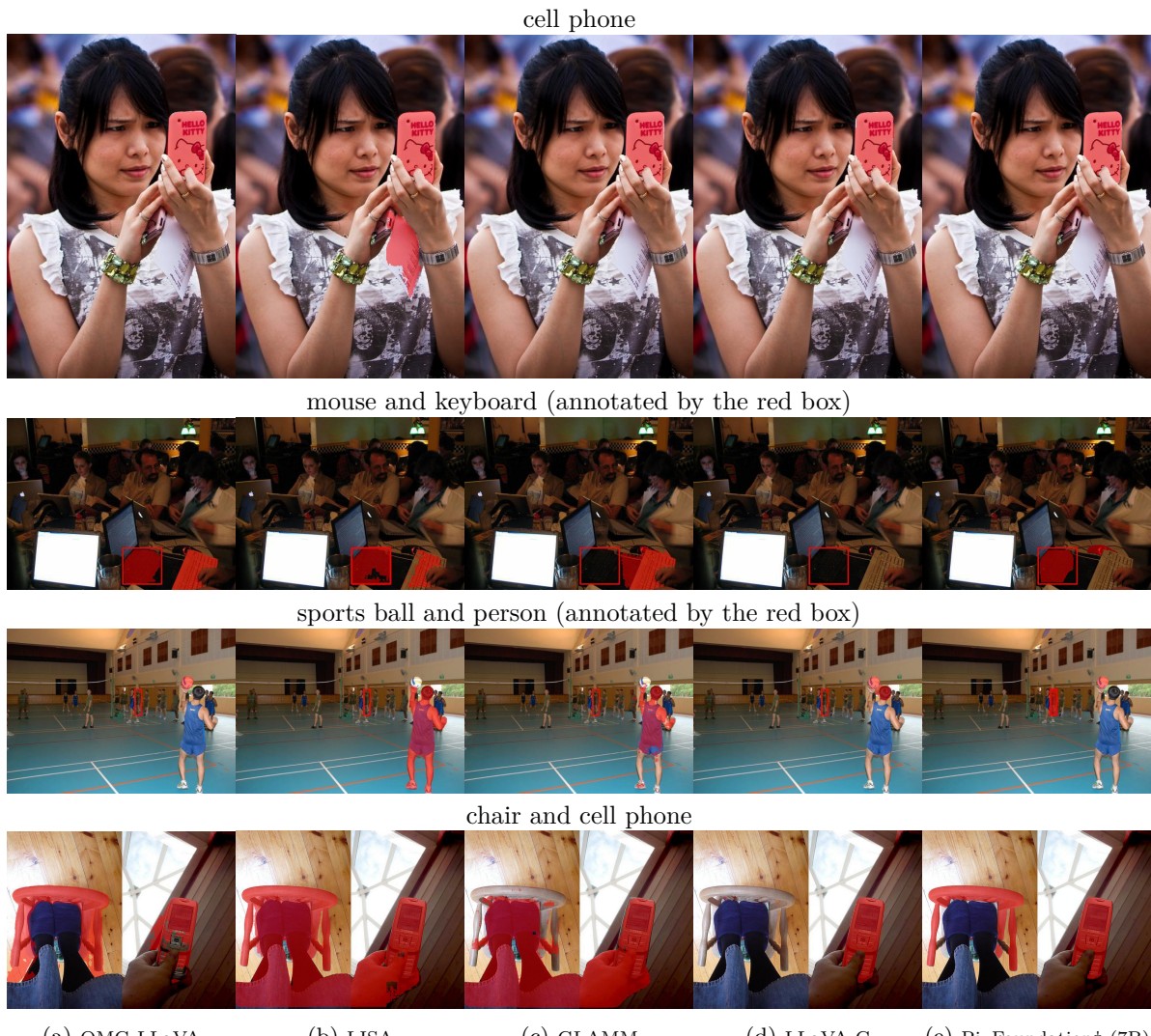

Figure S10: **PixCV-Bench qualitative comparison** between the pixel-level visual grounding following the second probing. The referred expression used in the segmentation is shown on top of each row. It shows similar to PixMMVP that mining for the grounding within MLLMs that were not trained with pixel-level grounding supervision paired with the oracle selection outperforms pixel-level MLLMs.

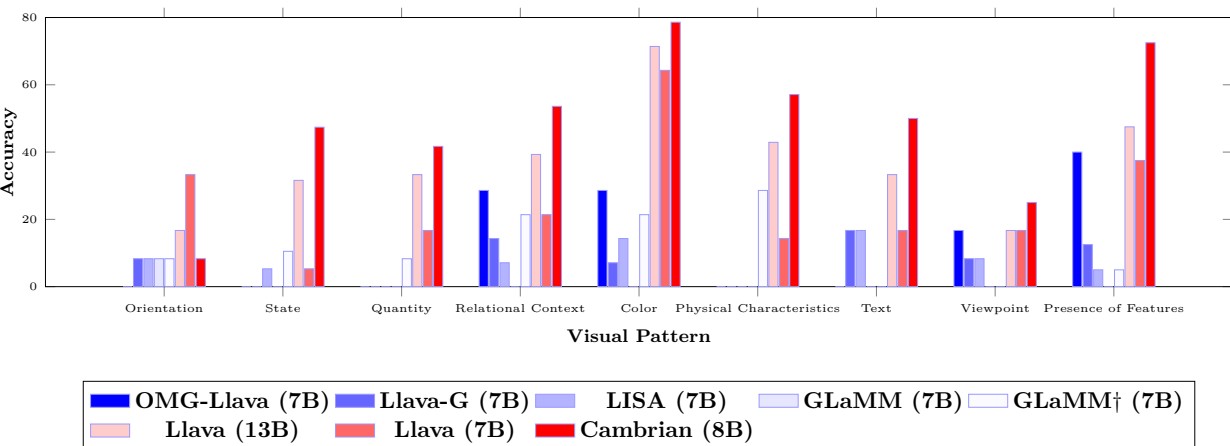

Figure S11: **PixMMVP fine-grained analysis** of the studied models performance across the different visual pattern, showing the model's accuracy with each pattern.

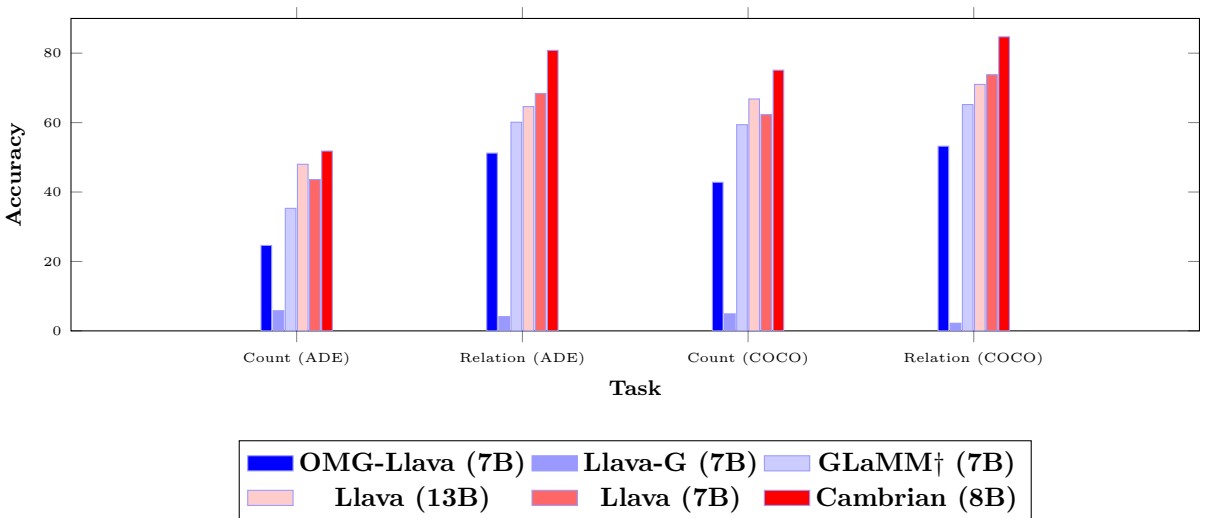

Figure S12: **PixCV-Bench fine-grained analysis** of the studied models performance across the different visual patterns in ADE20K and COCO, showing the model's accuracy with each pattern.

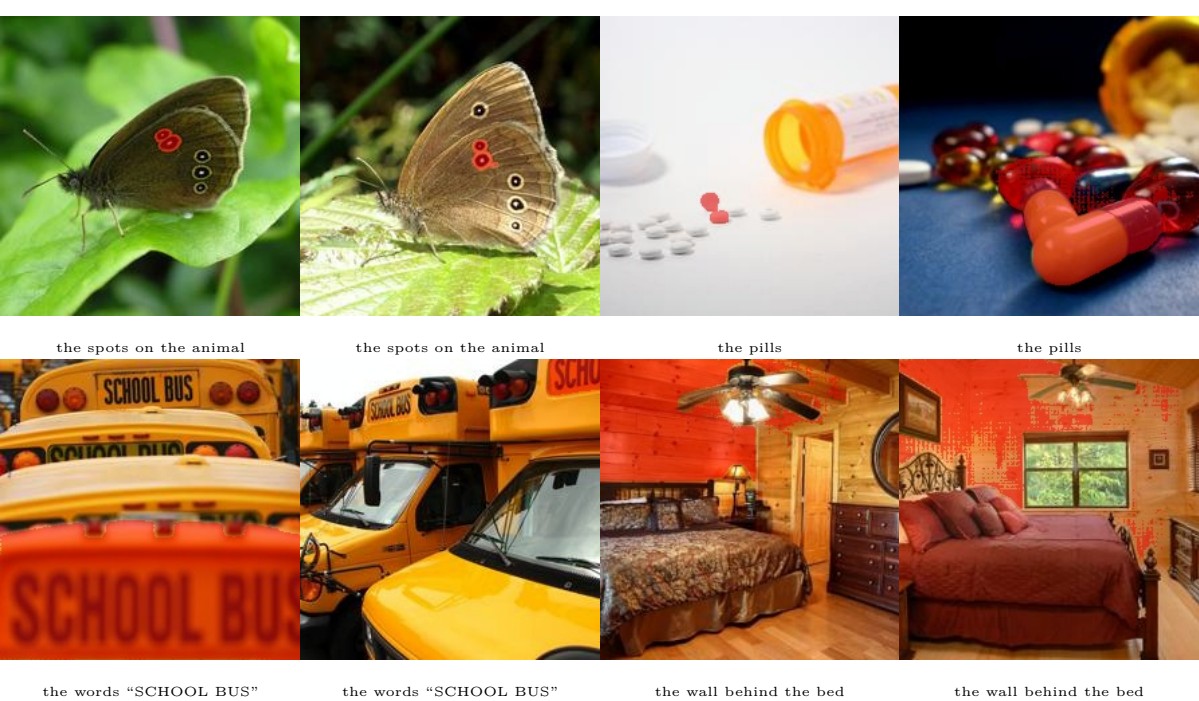

Figure S13: **Failures of the *oracle* upper bound,** PixFoundation†, using Cambrian (8B) as base MLLM on PixMMVP. It shows the failures mostly emerge in quantity or counting tasks. It also shows that the upper bound is inheriting SAM failures and the ambiguity arising in the referred expression itself, e.g., "the wall behind the bed", which direction does "behind" indicate.

