# OpenReview forum: "PixFoundation: Are We Heading in the Right Direction with Pixel-level Vision Foundation Models?"
_TMLR — Decision pending for TMLR_

### Review · Reviewer_1iJq · 2026-04-06

**Summary Of Contributions:**

This paper demonstrates that training Multi-modal Large Language Models (MLLMs) with explicit mask supervision degrades their core visual question answering (VQA) and language capabilities. To prove this, the authors introduce two paired benchmarks (PixMMVP and PixCV-Bench) for simultaneous VQA and grounding evaluation, alongside tailored prompt sensitivity analyses. Additionally, they introduce "PixFoundation", an interpretability tool revealing that visual grounding typically emerges from tokens describing an object's physical traits or location, rather than the exact noun phrase.

- Strengths: The paired evaluation framework is highly effective at exposing the "catastrophic forgetting" of language skills in specialized MLLMs. The interpretability finding regarding where grounding actually emerges within attention maps is a significant contribution to the field of AI interpretability. The authors also provide thorough baseline comparisons across multiple state-of-the-art models like Qwen2.5-VL, LLaVA 1.5, and Cambrian.

- Weaknesses: The proposed interpretability tool relies heavily on a closed-source model (GPT-5.1) for its automatic variant, which limits reproducibility, though an open-source ablation using Cambrian is provided in the appendix. Additionally, the upper bound "oracle" mechanism naturally inherits limitations from the underlying SAM model, such as struggling to segment multiple instances of the same object

**Additional Comments:**

This is an interesting with good premise. The shift towards paired evaluation (VQA + grounding) is exactly what the field needs to prevent benchmark gaming and ensure holistic model improvements. The visual examples provided in the figures clearly illustrate the interpretability claims and make the paper a compelling read.

**Audience:**

Yes

**Audience Explanation:**

The findings are highly relevant to researchers working on multi-modal foundation models, computer vision, and AI interpretability.

- The observation that explicit segmentation training degrades core language reasoning is a critical insight for model developers who are trying to unify vision and language tasks.

- Additionally, the novel analysis of attention maps showing that token grounding shifts to adjectives, states, or positional tokens rather than explicit nouns will be of great interest to the interpretability community.

**Claims And Evidence:**

Yes

**Claims Explanation:**

The authors provide extensive quantitative and qualitative evidence to support their claims.
- They evaluate five state-of-the-art pixel-level MLLMs (LISA, GLAMM, OMG-LLaVA, LLaVA-G, RGA) against standard MLLMs augmented with SAM.

- The results clearly show a 3-14% performance drop in VQA accuracy for pixel-level models compared to their standard baseline counterparts.

- The claims regarding prompt sensitivity are backed by rigorous testing using specific textual mutations (typos, paraphrasing) and novel visual cropping/masking techniques applied directly to the grounding task.

- Furthermore, the core claim about when grounding emerges is convincingly supported by location and concept histograms, proving that grounding frequently correlates with conceptual categories like "color and appearance" or "location and position" instead of the exact referring expression.

**Requested Changes:**

- Please clarify the exact implementation details (prompt) of the GPT-5.1 judge in the main text rather than strictly hiding it in the appendix, as it is a core component of the automatic PixFoundation pipeline.

- Please expand the failure case analysis to discuss potential strategies for overcoming the multi-instance segmentation failures inherited from the Segment Anything Model (SAM).

- Please include a deeper discussion on why certain architectures (like RGA) resist VQA degradation better than others (like GLAMM), as this could provide actionable insights for future model architectures

---

> ### Author Response · Authors · 2026-06-05
> **Rebuttal Response**
>
> We thank the reviewer for their thoughtful comments and positive feedback.
>
> > The proposed interpretability tool relies heavily on a closed-source model (GPT-5.1) for its automatic variant, which limits reproducibility, though an open-source ablation using Cambrian is provided in the appendix. Clarify the exact implementation details (prompt) of the GPT-5.1 judge in the main text rather than strictly hiding it in the appendix,
>
> We have modified Sec. 4.1 to include these in the main text, we also promise to release the code for using the closed-source model to ensure its reproducibility. While it might be harder to reproduce due to API access fees, yet we provide as mentioned the open source model results. Table S3 shows that even with a relatively old open source model, we are still capable of surpassing the naive (a+s) baseline.
>
> > The upper bound "oracle" mechanism naturally inherits limitations from the underlying SAM model, such as struggling to segment multiple instances of the same object. The failure case analysis to discuss potential strategies for overcoming the multi-instance segmentation failures inherited from the Segment Anything Model (SAM).
>
> We add the requested in Appendix Section D.2 with potential strategies to solve this issue.
>
> > a deeper discussion on why certain architectures (like RGA) resist VQA degradation better than others (like GLAMM),
>
> 📌 Note that our work is focused on the benchmarking efforts identifying the failures and the interpretability mechanism. The latter helped provide an off-the-shelf mechanism to perform visual grounding without degrading VQA abilities and also to study when grounding emerges in MLLMs with respect to the output tokens. Our aim was not to provide a new architecture/training recipes for pixel-level MLLMs. Nonetheless, we added in the appendix B4 and Table S4 results for two sets of models. **In each set we ablate with vs. without the segmentation decoder during inference and compare MLLMs with vs. without mask supervision (their counterpart backbone MLLM).** The discussion details the potential strategies why RGA is better, yet it is still suboptimal and we confirm this quantitatively. We leave future work to use some of our findings to drive better pixel-level MLLMs towards the direction of generalist models. These should perform well across different interconnected tasks in language and vision including the visual grounding task. While it is interesting to explore training recipes and new architectures, e.g., generating polygonal points in language w/o segmentation decoder, this requires a separate dedicated paper. Hence, it is out of the scope of our work and will broaden our scope beyond what a single paper can handle.

---

> > ### Comment · Reviewer_1iJq · 2026-06-08
> >
> > I appreciate that the GPT-5.1 implementation details have been integrated into the main text (Section 4.1) and that open-source alternatives were validated. Committing to releasing the code for the closed-source model pipeline also significantly improves the reproducibility of the PixFoundation framework.
> >
> > The added discussion in Appendix D.2 regarding multi-instance segmentation failures and the architectural ablation studies in Appendix B.4 adequately address my concerns about SAM's limitations and the performance retention of models like RGA. While a full architectural redesign is understandably out of scope for this benchmark-focused paper, the newly provided insights lay a solid foundation for future work in generalist multi-modal models.
> >
> > Given the timeliness of the paired evaluation framework and the valuable interpretability findings regarding when grounding emerges, I believe this paper makes a strong contribution to the community. I think my concerns have been addressed.

---

> > > ### Author Response · Authors · 2026-06-10
> > >
> > > We thank the reviewer for his positive feedback.

---

### Review · Reviewer_16pE · 2026-04-18

**Summary Of Contributions:**

The authors report that Multimodal large language models (MLLMs) exhibit limited ability in visual question answering (VQA). Several studies are then conducted around this topic. Two benchmarks are provided to evaluate both VQA and grounding. Other studies, such as prompt sensitivity analysis and explainability analysis.

**Audience:**

Yes

**Audience Explanation:**

The benchmarks provided might be interesting. The prompt sensitivity analysis, if done with more structured experiments, might be a good contribution. The analysis of this paper confirms current discussions that MLLMs might have a preference for the role of language in visual grounding. Regarding this, the authors might need to include studies on MLLM visual token pruning, which reflects the opposite side that the visual information might still need further enhancement (as the current bandwidth is not fulfilled by MLLM training).

**Broader Impact Concerns:**

NA.

**Claims And Evidence:**

No

**Claims Explanation:**

1. The paper is poorly written. The overall structure is a little bit confusing. The listed contributions are ranked, while several of them are marked as "very critical" (e.g., Furthermore, More importantly). The problem remains the same for the Intro writing. Multiple questions have arisen. The readers might get lost in these points. It would be great to reframe the current writing. Some sentences are too long (e.g., We identify an interesting connection between the identified output tokens and the output grounding from the attention maps that gives insights into how these models reason). Some discussions are in order that reviewers may find it hard to follow (e.g., Sec. 3.2).
2. Several critical questions take place in Sec. 3. First of all, the proposal benchmarks have limited samples (e.g., 300 images), which are designed to test pixel-level and standard MLLMs. Following concern 1, a natural question would be: are the new proposed benchmarks covering in-distribution pixel-level segments or out-of-distribution counterparts? Without discussing this question, it would be dangerous to provide the following analysis. Third, the prompt sensitivity evaluation lacks fundamental details. From my understanding, the provided discussions are case studies without systemetically exam the ability of the model, which brings critical concerns on the novelty and whether the current findings are interesting.
3. The interpretability mechanism is not novel, and the post-hoc explanation can not provide sufficient enough of support/contributions. This problem is twofold. First, the interpretability approach is highly reliant on the previous approach; its own novelty remains questionable. Second, how to create such an explanation is engineering-heavy. For example, the authors extract the maximum attention point, prompt SAM with it, highlight the generated mask in the image, and ask the MLLM to select the highlighted image that best describes the queried referring expression from all the masks. Whether the explanation is provided by SAM/prompt engineering/MLLM is questionable. Also, the discussion on explainability is not sufficient, covering only post-hoc explainability [1-3].

[1] Deepncm: Deep nearest class mean classifiers

[2] Visual recognition with deep nearest centroids

[3] Prototype-based semantic segmentation

**Requested Changes:**

Please see my concerns above. The experimental design and claims of this paper are not sufficient.

---

> ### Author Response · Authors · 2026-04-20
> **Clarifications before the rebuttal period**
>
> We thank the reviewer for his comments. Once the rebuttal phase is open, we will provide a point-by-point response in another follow-up comment. However, would it be possible if the reviewer provides a clear list of requested changes, w.r.t, the poor writing comment and , before this phase? It seems currently too vague concerns for us to address:
> - The overall structure is a little bit confusing. (What is confusing in the structure? What would the reviewer want to see to remove this confusion?)
> - The listed contributions are ranked, while several of them are marked as "very critical" (e.g., Furthermore, More importantly). (They are enumerated. "More importantly" was referring to how visual grounding methods emerged without doing the proper comparison effort to the simple baseline of MLLMs without masks + SAM. That is a major finding indeed discussing pitfalls that previous literature fell into, which simply emphasizes the previous finding of VQA's limited abilities, hence. Where is the issue?)
> - Some sentences are too long (2-line sentences are standard and not referred to as long. According to ChatGPT, it isn't a long sentence, which was trained with broad datasets of text. Maybe the reviewer can clarify. More how is this long and how does it relate to poor writing?)
> - Some discussions are in order that reviewers may find it hard to follow (What exactly is hard to follow in Sec. 3.2. w.r.t the order? If the reviewer can provide what he did not understand from that section order it will help us understand what change to implement)
>
> - Also, can the reviewer clarify his question, "are the new proposed benchmarks covering in-distribution pixel-level segments or out-of-distribution counterparts?" Do they mean in-distribution w.r.t refCOCO or other visual grounding training data?
>
> Thank you.

---

> > ### Comment · Action_Editor_BBTV · 2026-05-09
> >
> > Dear Reviewer 16pE,
> >
> > Thank you for your comments. As requested by the authors, would you mind elaborating on your review and providing a few more details on your main concerns, so that the authors can address them more effectively?
> >
> > Thanks,
> >
> > AE

---

> > > ### Comment · Reviewer_16pE · 2026-05-11
> > >
> > > Thank you for the follow-up comments. I also would love to see the authors provide additional technical detail in the point-by-point response.
> > >
> > > 1. Regarding the structure confusion, the major confusion lies in the introduction. 1. The 2nd paragraph is confusing due to the overall claim structure. Specifically, the authors first claimed that "these benchmarks did not evaluate the recent pixel-level ones," which leads to the 1st question (but the authors show in the following that other benchmarks have related studies). This step-by-step narrative can easily confuse the reviewers. The best way to do it is to quickly discuss the current insufficiency of pixel-level study): “Are the current pixel-level MLLMs trained with mask supervision heading
> > > in the right direction to improve both grounding and visual question answering (VQA)?” While the authors' findings confirmed that "the majority of pixel-level MLLMs still fall short in such a challenging setting." The finding, however, is redundant as it has been observed by concurrent work (this is fine, but the core question remains ambiguous after the long discussions. I suggest that the authors focus on a critical research direction; the related discussions can be placed in the related work section.
> > > 2. In Sec 3.2, the discussions on prompt sensitivity evaluation lack fundamental details, without design, quantitative numbers. I am concerned that this paper falls more into the realm of case studies than into a scientific paper. In Sec 3.3, my concern remains, as "additional mask selection", "maximum attention point" are new concepts without further discussions or notations. Other small problems, like "we consider that the point prompt ambiguity is crucial in identifying fine-grained details," lack more discussion of the motivation and intention behind it.
> > > 3. For in-distribution related discussion, yes, I mean in-distribution w.r.t refCOCO.

---

> > > > ### Author Response · Authors · 2026-05-18
> > > > **Thanks**
> > > >
> > > > Thanks for the clarification. I am awaiting the third review to respond across the three for uniformity.

---

> ### Author Response · Authors · 2026-06-05
> **Rebuttal Response**
>
> > 1.1 - The paper is poorly written.
>
> We believe our writing is of high quality, given that all reviewers understood the work clearly. We acknowledge the reviewer's concerns regarding certain aspects of the writing that we addressed in the manuscript. These do not amount to poor writing, **but rather minor edits.**
>
> > 1.2 - structure confusion...authors claimed that "these benchmarks did not evaluate the recent pixel-level ones,"…but the authors show in the following that other benchmarks have related studies.
>
> 📌 We have modified Sec. 1 to clarify that the vision-centric VQA benchmarks carefully designed w.r.t what MLLMs fail at (i.e., MMVP & CV-Bench), have not evaluated pixel-level MLLMs. Hence, our first question emerges with emphasis on these challenging vision-centric benchmarks and extending it to the pixel-level tasks. Our concurrent work did not tailor pixel-level extensions of these VQA datasets. **We also clarify further our novelty w.r.t this work: (i) the paired evaluation, (ii) the language and visual prompt sensitivity evaluation in grounding and (iii) the interpretability mechanism.**
>
> > 1.3 - "the majority of pixel-level MLLMs still fall short in such a challenging setting." The finding, however, is redundant
>
> 📌 We have revised the first question (Sec. 1) **to emphasize the MLLMs' robustness to prompt variations**. We also show failures of well-known general-purpose MLLMs, Qwen2.5-VL, in relation to these variations. *This is crucial since previous literature and a previous review of our work tried to overlook the impact of language on visual grounding.* Finally, we show the results for the non-unified baselines, base MLLM + SAM, to standardize that practice. **These three are not redundant.**
>
> > 1.4 - Some sentences are too long.
>
> We fixed this.
>
> > 2.1 - The proposal benchmarks have limited samples (e.g., 300 images).
>
> 📌 Our novel benchmarks (PixMMVP & PixCV-Bench) provide in total 1,738 images, which is more than well established benchmarks. In comparison, RefCOCO variants provide 750-1500 images val/test sets. RefCOCO started to saturate, e.g., for referring detection (acc) are at 90%+ and for referring segmentation (iou) are catching up near 80%. Our benchmarks are constructed based on MLLMs' failures that makes them challenging, which is reflected in our results near 40-50%.
>
> > 2.2 - in-distribution pixel-level segments or out-of-distribution counterparts
>
> 📌 We added an analysis of the ID/OOD using TSNE plots between source, reference and target datasets, where the source and reference are ID and target dataset is the one in question. We show our splits, PixMMVP, PixCV-Bench (ADE) and PixCV-bench (COCO) w.r.t. RefCOCO six variants in Appendix C.1. Figure S2 shows the first two splits are more OOD w.r.t RefCOCO.
>
> > 2.3 - In Sec 3.2, the discussions on prompt sensitivity evaluation lack fundamental details, without design, quantitative numbers.
>
> We have revised Sec.3.2 to include more details. We also provide systematic evaluation of the visual prompts variation with respect to the control parameters as a motivation.
>
> > 2.4 - "additional mask selection", "maximum attention point" are new concepts without further discussions or notations.
>
> Sec. 3.3 is revised.
>
> > 2.5 - "we consider that the point prompt ambiguity is crucial in identifying fine-grained details," lack more discussion of the motivation.
>
> It is well known that SAM can accommodate point prompt ambiguity, dorsal fin vs. shark in Fig. 8. Fig. 7a ablates PixFoundation vs. (a+s) using the point accuracy, which ensures the gain is from, when grounding emerges, rather than accommodating the ambiguity.
>
> > 3.1 - the interpretability approach is highly reliant on the previous approach
>
> 📌 While our interpretability mechanism is simple, the insight is significant, as noted by other reviewers. *Some of the most powerful interpretability tools such as CAM, are based on simple empirical ideas that provide meaningful insights; this does not reduce our work.* **The considerable gains from PixF vs. (a+s), w/ 20-30%, show the importance of our insight (Tab. 2, 3 & Fig. 7a).** The mask selection using MLLM as a judge is novel on its own. Our insight is being used by concurrent works to improve their method, e.g., Wu et. al. After the release of our work, Feb, 6, 2025, they were inspired to use it to develop a keyword selector part of their revised approach March, 23, 2025, they were aware of our work before its release.
>
> > 3.2 - Whether the explanation is provided by SAM/prompt engineering/MLLM is questionable...engineering-heavy
>
> Our method is simple and has no hyperparameters to tune. We provide the Point Accuracy, Fig. 7a, which shows accuracy w/o SAM. We also outperform (a+s) independent of MLLM; our results w/ closed source (Tab. 2&3) or open source (Tab.S2).
>
> > 3.3 - the discussion on explainability is not sufficient
>
> We add it in Sec 2 related work.
>
> >  MLLM visual token pruning
>
> This is out of our scope, so we left it out.

---

### Review · Reviewer_8CR8 · 2026-05-23

**Summary Of Contributions:**

The authors examine whether current pixel-level MLLMs can effectively improve both visual question answering and pixel-level grounding.
The paper makes two main contributions: it introduces two paired benchmarks, PixMMVP and PixCV-Bench, for evaluating VQA and segmentation on the same object of interest, and it proposes PixFoundation, an attention-mining pipeline for analyzing when grounding emerges in standard MLLMs.
One interesting finding is that grounding may arise not only from the exact referring expression, but also from output tokens describing object parts, appearance, location, context, or state.

A key strength of the work is its timely diagnostic perspective.
The paired benchmark design is useful, the comparison across pixel-level and standard MLLMs is informative, and the prompt sensitivity analysis highlights an important issue: some pixel-level MLLMs trained with mask supervision may suffer from degraded VQA, instruction-following, or robustness.

At the same time, the paper could benefit from a more carefully scoped interpretation of its results.
The meaning of “right direction” is not clearly defined, which makes the framing broader than what the experiments directly support.
In addition, while the experiments reveal important failure cases, they do not fully disentangle whether these failures arise from pixel-level supervision, training data, model scale, decoder design, or prompt mismatch.
Finally, PixFoundation relies on a multi-stage pipeline involving SAM and mask selection, making it difficult to compare directly with unified pixel-level MLLMs.
Overall, the paper offers valuable benchmarks and observations, but its broader conclusions would be stronger if presented with a more cautious and nuanced framing.

**Additional Comments:**

I would like to note that I am not a specialist in this exact area, but I have done my best to provide a careful and constructive review.
I hope my comments are helpful for improving the work, and I am open to further discussion or clarification from the authors.

**Audience:**

Yes

**Audience Explanation:**

Although I have raised several concerns, the paper also has clear strengths that may be of interest to the TMLR audience.
It tackles an important question: whether pixel-level MLLMs can jointly support visual question answering and fine-grained grounding.
The paired benchmark design is particularly valuable, as PixMMVP and PixCV-Bench evaluate VQA and segmentation for the same object of interest, offering a more informative diagnostic setting than evaluating these abilities separately. \
In addition, the PixFoundation analysis provides an interesting interpretability perspective, suggesting that standard MLLMs may already encode latent grounding signals that can be mined and further studied. This insight could motivate future work on model design and analysis tools.

**Broader Impact Concerns:**

I don't have any concerns related to this.

**Claims And Evidence:**

No

**Claims Explanation:**

- **W1: The central "right direction" framing is underdefined and broader than the evidence supports.**
The paper asks whether current pixel-level MLLMs are heading in the "right direction," but it does not clearly define what this direction means.
Based on the experiments, the intended criterion seems to be that a pixel-level MLLM should improve grounding while preserving VQA ability, instruction following, language robustness, and general multimodal reasoning.
However, this criterion is not explicitly formalized or separated into measurable desiderata.
As a result, the framing feels somewhat overstated: the experiments show that several current pixel-level MLLMs struggle on the proposed paired VQA-grounding benchmarks, but this does not necessarily imply that the broader research direction is wrong.
A more precise claim would be that some existing mask-supervised pixel-level MLLMs exhibit unfavorable tradeoffs between grounding and language/VQA ability under challenging paired evaluation.

- **W2: The experiments reveal important failures but do not isolate their causes.**
The proposed paired benchmarks are useful because they jointly evaluate whether a model can answer a visual question and ground the relevant visual evidence.
However, the experiments are mainly diagnostic rather than causal. They show that some pixel-level MLLMs fail in VQA, grounding, instruction following, or robustness, but they do not identify why these failures occur.
Degraded VQA performance could result from many factors, including pixel-level supervision, specialized decoder design, limited instruction-tuning data, model scale, training recipe, prompt format, and so on.
Without controlled comparisons using the same base MLLM with and without mask supervision, or the same architecture under different training mixtures, it is difficult to attribute the failures specifically to pixel-level supervision or to the general direction of pixel-level MLLM design.

- **W3: The PixFoundation comparison is informative but not fully fair as a direct model comparison.**
PixFoundation provides an interesting interpretability insight by showing that grounding signals can emerge from output tokens related to appearance, location, object parts, context, or state, rather than only from the exact referring expression.
However, using PixFoundation as a comparative baseline against unified pixel-level MLLMs raises fairness concerns.
PixFoundation is a multi-stage pipeline relying on MLLM attention mining, noun-phrase extraction, SAM-based candidate mask generation, and automatic or oracle mask selection.
PixFoundation$^{\dagger}$ is especially not a deployable baseline because it uses ground-truth masks for oracle selection.
Therefore, the comparison is not simply between standard MLLMs and pixel-level MLLMs, but between unified pixel-level MLLMs and a post-hoc pipeline composed of multiple strong components.
The results are better interpreted as evidence that standard MLLMs contain latent grounding signals that can be extracted with additional tools, rather than as direct evidence that unified pixel-level MLLMs are inferior.

**Requested Changes:**

- **C1. Clearly define and narrow the "right direction" claim.**
The authors could explicitly define what they mean by a desirable direction for pixel-level MLLMs, such as improving grounding while preserving VQA and general-purpose language ability.
They could then reframe the paper as a diagnostic study of current VQA–grounding tradeoffs, rather than implying that the entire research direction is wrong.

- **C2. Add controlled experiments to isolate the source of failures.**
To strengthen the causal interpretation, the authors could compare the same base MLLM under controlled variants: with and without mask supervision, with and without a segmentation decoder, or with different instruction-tuning mixtures.
This would help determine whether the observed degradation comes from pixel-level supervision itself, training data imbalance, model capacity, decoder design, or prompt/task mismatch.

- **C3. Reposition PixFoundation as an interpretability and upper-bound tool, not a direct competitor.**
The authors could more clearly separate deployable baselines from oracle or multi-stage analyses.
For example, the authors could report PixFoundation$^{\dagger}$ only as an upper bound, emphasize automatic PixFoundation separately, and add runtime/component ablations for SAM, mask selection, and noun-phrase extraction.
This would make the comparison with unified pixel-level MLLMs more balanced and fair.

---

> ### Author Response · Authors · 2026-06-05
> **Rebuttal Response**
>
> We thank the reviewer for their positive comments and actionable items.
>
> > The meaning of “right direction” is not clearly defined… measurable desiderata... The paper asks whether current pixel-level MLLMs are heading in the "right direction," but it does not clearly define what this direction means. (C1).
>
> We rewrite our first research question in Sec. 1 to reflect our emphasis on improving VQA and visual grounding simultaneously in both accuracy and robustness to prompt variations. Reframed to: *“Are the current pixel-level MLLMs trained with mask supervision heading in the right direction to improve both grounding and visual question answering (VQA) while maintaining robustness to prompt variations?”*. We also motivate the reason we believe this is the right direction, as we are shifting more towards generalist models that are performant in various tasks. Nonetheless, we clearly define our scope as the **paired evaluation of visual grounding and VQA accuracy and robustness only**, which is a step towards the aforementioned.
> Sec. 4.2 states we answer this question through evaluating both VQA and visual grounding. Sec. 4.1 details the **measurable desiderata in terms of evaluation protocols and metrics.**
>
> > the framing feels somewhat overstated: the experiments show that several current pixel-level MLLMs struggle ... this does not necessarily imply that the broader research direction is wrong.
>
> We are not claiming the broader direction of building pixel-level MLLMs as a wrong direction. This is already conveyed in Sec. 3.2 & 3.3, *“The goal is to push these models in the direction of learning visual grounding abilities on the pixel-level without sacrificing the language abilities that are integral to grounding.”*, *"there could be better pixel-level MLLMs built without the loss of their original language and grounding capabilities"*.
>
> We modify Sec. 1 to highlight it. **We believe the wrong direction is focusing solely on localization/segmentation while overlooking the models’ language abilities, which are interconnected to their visual grounding abilities.** This is confirmed with Qwen2.5-VL examples in Fig. 4.
>
> > they do not fully disentangle whether these failures arise from pixel-level supervision, training data, model scale, decoder design, or prompt mismatch...C2. authors could compare... with and without mask supervision, with and without a segmentation decoder, or with different instruction-tuning mixtures..
>
> 📌 Note that our work is focused on building the dataset, the benchmarking and analysis effort, identifying the failures and the interpretability mechanism. The latter helped provide an off-the-shelf mechanism to perform visual grounding without degrading VQA abilities and also to study when grounding emerges in MLLMs with respect to the output tokens. Our aim was not to provide a new architecture/training recipes for pixel-level MLLMs. Nonetheless, we added in Appendix B4 results for the variants summarized in the table below. Note that it's out of the scope of our work to train for mask supervision without the segmentation decoder, as this entails a completely new design and model that requires separate dedicated work.
>
> | Method  |  w/ Mask Supervision | w/ Seg. Decoder  |
> |---|---|---|
> | RGA-Full  | &#x2713;  | &#x2713;  |
> | RGA-Part  | &#x2713;  |  &#x2717; |
> | Qwen2.5-VL  | &#x2717;  |  &#x2717; |
> | LISA-Full  |  &#x2713; |  &#x2713; |
> | LISA-Part  | &#x2713;  | &#x2717; |
> | LLaVA-1  |  &#x2717; | &#x2717; |
>
> The discussion isolates some of the failure reasons and discusses the potential strategies for why RGA is better, yet RGA is still suboptimal. We leave future work to explore instruction tuning mixes and architectures using some of our findings to drive better pixel-level MLLMs towards the direction of generalist models. While it is interesting to explore training recipes and new architectures, e.g., generating polygonal points in language w/o segmentation decoder, this again requires a separate dedicated paper.
>
> > Finally, PixFoundation relies on a multi-stage pipeline involving SAM and mask selection, making it difficult to compare directly with unified pixel-level MLLMs… C3. Reposition PixFoundation as an interpretability and upper-bound tool, not a direct competitor
>
> 📌 We clarify that unified pixel-level MLLMs do incorporate SAM or alike in Sec. 4.1 "These were either equipped with a SAM or SAM 2 decoder or with other powerful decoders, e.g., OMG-Seg. " Some also train on SA-1B data for grounding. Already our positioning is that it's an interpretability tool in Sec.1, 3.3 and 4 (Table 2 - 2nd col.). We use an additional mask selection, but it's our contribution. We can still compare quantitatively to unified models for reference to question the underlying lang./grounding capabilities that were lost. We also add our simple non-unified baselines for fairness.
>
> > runtime/component ablations for SAM, mask selection, and noun-phrase extraction
>
> We add these in Appendix B.5.

---

> > ### Comment · Reviewer_8CR8 · 2026-06-20
> >
> > Thank you to the authors for their efforts in providing additional descriptions and details.
> > In particular, the added ablation studies for the MLLMs were very helpful.
> > I believe that most of my concerns have now been adequately addressed.

---

> > > ### Author Response · Authors · 2026-06-21
> > >
> > > We thank the reviewer for their positive feedback.

---

### Author Response · Authors · 2026-06-05

We thank the reviewers for the detailed and constructive feedback. We start with reiterating their positive comments highlighting the strength of our work beyond minor additional experiments and edits.

> A key strength of the work is its **timely diagnostic** perspective.

> The **paired benchmark design is useful**, the comparison across pixel-level and standard MLLMs is informative,

> the **prompt sensitivity analysis highlights an important issue**

>The paired evaluation framework is highly effective at **exposing the "catastrophic forgetting" of language skills** in specialized MLLMs.

> The interpretability finding regarding where grounding actually emerges within attention maps is **a significant contribution to the field of AI interpretability.**

> The authors also provide **thorough baseline comparisons** across multiple state-of-the-art models like Qwen2.5-VL, LLaVA 1.5, and Cambrian.

Our experiments are strong in their breadth and depth and they were augmented further as part of this rebuttal. We believe the TMLR and the ML community will benefit from our work that does not shy out from providing a detailed analysis of previous works in pixel-level MLLMs. In addition to the significant insights from our interpretability mechanism, where the significance is highlighted in terms of quantitative results. **It is also highlighted in the contribution to a responsible AI approach that favours interpretability and robustness over mere bold numbers.**

*We further provide a detailed point-by-point response paired with the revised manuscript, where all the major changes are highlighted in blue.*